# Last interglacial sea levels within the Gulf of Mexico and northwestern Caribbean Sea

Alexander R. Simms[1]

[1]Department of Earth Science, University of California Santa Barbara, Santa Barbara, CA 93106, USA
*Correspondence to*: Alexander R. Simms (asimms@geol.ucsb.edu)

**Abstract.** During the Last Interglacial (LIG) the volume of additional water in the world's oceans was large enough to raise "global" sea levels about 6-9 m higher than present levels.  However, LIG sea levels vary regionally and those regional differences hold clues about the past distribution of ice sheets and local rates of subsidence and tectonic uplift.  In this study, I used a standardized database template to review and summarize the existing constraints on LIG sea levels across the northern Gulf of Mexico and Caribbean shoreline of the Yucatan Peninsula.  In total, I extracted 32 sea-level indicators including the insertion of 16 U-series ages on corals, 1 electron spin resonance age, 2 amino acid racemization ages and 26 luminescence ages. Most dated sea-level indicators for the northern Gulf of Mexico are based on optically stimulated luminescence (OSL) ages of beach deposits of a mappable LIG shoreline.  This shoreline extends from the Florida Panhandle through south Texas but is buried or removed by the Mississippi River across most of Louisiana.  A similar feature is observed in satellite images south of the Rio Grande within the Mexican portions of the Gulf of Mexico but has yet to be dated.  Elevations measured on portions of this feature close to the modern coast point to sea levels less than 1 m to ~5 m higher than present for much of the northern Gulf of Mexico.  However, a few, albeit undated, portions of the same shoreline located at more inland locations point to sea levels up to +7.2 m attesting to up to 7 m of differential subsidence between the inland and coastal sites.

Across the Yucatan Peninsula, U-series dating of corals has provided the main index points for LIG sea levels.  Other carbonate coastal features such as beach ridges and eolianites have also been described, but rely on corals for their dating.  The maximum elevation of the LIG coral-based relative sea-level (RSL) estimates decrease from around +6 m across the Caribbean shoreline of the Yucatan Peninsula near Cancun, Mexico to as low as -6 m to the south beneath the southern atolls of Belize, although discussion continues as to the validity of the ages for these southern corals.  If these lower elevations corals are LIG in age, their below-present elevations may be a result of vertical motion along faults dipping into the Cayman Trough.  South of Belize only one purported LIG coral has been dated on the Isla de Roatán off the coast of Honduras at a likely tectonically uplifted elevation of 37.2 m.  Thus the elevation of LIG sea levels within the inland siliciclastic shorelines of Guatemala and Honduras as well as the southwestern Gulf of Mexico remain poorly constrained and potential venues for future research.

The database described in this paper is available open-access in spreadsheet format as Simms (2020), at this link: https://doi.org/10.5281/zenodo.4556163.

**1 Introduction**

During the Last Interglacial (LIG) Earth experienced global sea-surface temperatures on average 0.7±0.6 °C warmer than today (McKay et al., 2011) with global average sea levels reaching 6-9 m higher than current levels (Kopp et al., 2009; Dutton et al., 2015). As such it provides a possible analogue for what the future may bring to global coastlines (Dutton and Lambeck, 2012). However, debate continues as to the origin of the 6-9 m of excess meltwater with Greenland and West Antarctica as the two most likely candidates (Dutton et al., 2015). One method commonly applied to determining the hemispheric origin

for meltwater during the past is "sea-level fingerprinting" (Clark et al., 2002). "Sea-level fingerprinting" works on the premise that glacial-isostatic adjustment (GIA) results in differences in relative sea-level histories at different locations across the globe (Ferrel and Clark, 1976; Hay et al., 2014). The nature of this variability is a function of the past distribution of ice sheets and can thus be used for determining the origin of past meltwater contributions (Clark et al., 2002). In addition, the elevation of sea levels during the LIG are also commonly used as a datum for determining rates of tectonic uplift and loading-induced

subsidence (e.g. Simms et al., 2016; Paine, 1993). However, due to the regional variability caused in part by GIA, relative sea-level reconstructions are needed on a local basis for their use in determining vertical motions of the land.

This work is part of the World Atlas of Last Interglacial Shorelines (WALIS), a community effort to construct a database of LIG relative sea-level indicators (https://warmcoasts.eu/world-atlas.html). Within WALIS, this paper aims to summarize the

current knowledge on LIG shorelines and RSLs for the Gulf of Mexico and far western Caribbean. This summary includes data collected on LIG shorelines from the Panhandle of Florida to the coast of Honduras (Fig. 1). In total, this review covers 62 papers and 32 sites with constraints on paleo-sea levels. This included the insertion of 16 U-series ages on corals, 1 electron spin resonance age (ESR), 2 amino acid racemization ages (AAR), and 26 optically stimulated luminescence ages (OSL) into the WALIS database. The database for the region mentioned above is available open-access in spreadsheet format as Simms

(2020), at this link: https://doi.org/10.5281/zenodo.4556163. Database field descriptors are available from Rovere et al. (2020) at this link: https://doi.org/10.5281/zenodo.3961543. The following paragraphs give an overview of the geologic and literature background, present the datapoints included in the database and discuss the main opportunities for future work.

**2. Background**

**2.1 Geologic Overview**

The region of interest covered by this review contains two contrasting types of coastlines with respect to depositional settings and climate at the present and by extension during the LIG. Along the northern and western Gulf of Mexico, the coastline is dominantly siliciclastic with the LIG paleoshorelines marked by sandy paleo raised beaches and potentially paleo-barrier islands (Price, 1933; Otvos, 1972; 2005). Across the northern Gulf of Mexico, tides are semidiurnal and microtidal with tidal

ranges generally 0.4 to 0.6 m but approaching 1 m within the Florida Panhandle (Stumpf and Haines, 1998; Livsey and Simms, 2013). Wave energy is generally low with wave heights averaging less than 1 m (Hwang et al., 1998) with the exception of during the passage of tropical cyclones, which enter the Gulf of Mexico on average every 1.6 years (Parisi and Lund, 2008), and winter cold fronts originating from North America. The climate across the northern Gulf of Mexico varies from arid to semi-arid along the US-Mexico border to humid temperate conditions along the eastern Gulf of Mexico (Thornthwaite, 1948).


The northern Gulf of Mexico is a passive margin but locally, Quaternary growth faults have been identified (Yeager et al., 2019). However, most known active growth faults are located seaward of the LIG shoreline, although anthropogenic activities such as groundwater and hydrocarbon withdrawal have potentially contributed to motion along faults landward of the modern shoreline (White et al., 1997; Qu et al., 2015). The passive margin is experiencing subsidence across most of the coastline.

Attempts to quantify long-term subsidence usually rely on the elevation of the LIG (Paine, 1993; Simms et al., 2013) and are on the order of 0.05 mm/yr with locally higher rates around large deltaic centers (Dokka et al., 2006). Similar to motion along growth faults, groundwater and hydrocarbon extraction have led to increased rates of subsidence during historical times (Morton et al., 2006). Compared to the Texas and Louisiana coasts, less is known about the vertical motions of the Mississippi, Alabama, and Florida coasts. Few faults have been mapped although the area is not free of seismicity (Gomberg and Wolf,

1999). Otvos (1981) suggests the coast is uplifting very slowly over the Quaternary but estimates for the rate of uplift are lacking.

Farther to the south along the Yucatan Peninsula and south to Honduras, the coastline is marked by increasingly more tropical climates as well as a mixed siliciclastic/carbonate coastline with LIG shorelines largely marked by fossil coral reefs and

eolianites (Ward and Bradley, 1979; Gischler and Hudson, 1998; Blanchon et al., 2009). The coastline is more carbonate-dominated at the northeastern tip of the Yucatan Peninsula with an ever increasing influence of siliciclastics to the south such that within Honduras the carbonate environments are restricted to the offshore islands of Roatoan and the Swan Islands (Fig. 1). This change is a reflection of the Maya Mountains and Interior Highlands that hug the shore across portions of southern Belize and Honduras, respectively, as the coastline leaves the stable platform of the Yucatan Peninsula and nears the orogenic

belts marking the North American – Caribbean plate boundary (Pindell and Barrett, 1990). To the north, for most of its extent, the Gulf of Mexico and northern Yucatan shorelines are located along a passive margin except near the southwestern corner of the Gulf of Mexico where the Trans-Mexican Volcanic belt butts up against the Gulf (Ortega-Gutierrez et al., 1992). Like the northern Gulf of Mexico, the tidal range is relatively low, with values ranging from 0.1 m to 0.8 m (Rankey et al., 2021). Tradewinds drive the dominant wind direction from the northeast with average offshore wave heights of ~2.5 m (Rankey et

al., 2021). Similar to the northern Gulf of Mexico, wave heights increase during the passage of cold front, termed Nortes within Mexico, and tropical cyclones (Ojeda et al., 2017).

## 2.2 Literature Overview

Across the northern Gulf of Mexico, Price (1933) was one of the first to recognize the raised beach/barrier island shoreline across Texas now recognized as the Sangamon (LIG) shoreline. MacNeil (1949) mapped similar Pleistocene raised beach/barrier islands across western Florida. However, MacNeil (1949) separated features of similar age into more than one stage of Pleistocene shoreline development and potentially mixed others (Otvos, 1995). Over the next several decades discussion continued over the age of the raised beach/barrier island features mapped by Price (1933) and MacNiel (1949) with some suggestions of a mid-Wisconsin (~marine isotope stage (MIS) 3) age (Bernard and LecBlanc, 1965; Lundelius, 1972; Wilkinson et al., 1975; Shideler, 1986), a Sangamon (LIG) age (Price, 1933; Shepard and Moore, 1955; Price, 1958; Otvos, 1972a,b; Winker and Howard, 1977; Morton and Price, 1987; Tanner and Donoghue, 1992; Paine, 1993; Otvos, 1995), and possibly even portions of it dating to the middle Holocene (Donoghue and White, 1995; Blum et al., 2002; 2003). Some of this confusion may have stemmed from stratigraphic correlations with similar Pleistocene shallow-marine and beach deposits in the subsurface (Wilkinson, 1975; Shideler, 1986). Early attempts to obtain absolute ages of the feature relied on radiocarbon dating (Schnable and Goodell, 1968; Shideler, 1986; Otvos and Howat, 1996) but produced near finite ages adding to the confusion. However by the early 2000's a LIG origin was largely agreed upon thanks to a pair of studies by Blum et al. (2003) and Otvos (2005b) who provided the first absolute ages consistent with a LIG origin. Using OSL, they obtained ages varying between 116.1 ka and 137.8 ka, along the Mississippi and Alabama portions of the paleoshoreline but not without a few other spurious thermoluminescence ages. Burdette et al. (2012) and Simms et al. (2013) soon added additional OSL ages supporting a LIG origin to the feature in Florida and Texas, respectively.

Across the Yucatan, most early work on the carbonate shorelines was conducted by graduate students of Prof. J.L. Wilson of Rice University in the 1960s and 1970s (Ward, 1985). Purdy (1974) was one of the first to describe *in situ* corals of *Acropora palamata* within pre-Holocene limestone reefs of present-day Belize. These studies along with the work of a handful of other groups led to several publications discussing the sedimentology and diagenesis of the Yucatan's late Pleistocene calcarenites (Ward and Brady, 1976), strandlines (Ward and Brady, 1979; Lauderdale et al., 1979) and reefs (Purdy, 1974; Tebbutt, 1975; Ward and Halley, 1985), but ages on these rocks were first conducted by Szabo et al. (1978) within Mexico and Gischler et al. (2000) within Belize. Following the work of Jordan-Dahlgren (1997), Blanchon et al. (2009) and Blanchon (2010) examined a particularly well exposed section of the Pleistocene reefs within a theme park near Playa del Carmen, Mexico. Blanchon et al. (2009) combined detailed stratigraphic sections through the paleoreef within the park with new U-Th ages to place tight constraints on sea levels during the LIG. With this framework, they argued for two distinct levels of higher-than-present RSLs within the LIG. Mazzullo (2006) added a handful of additional LIG age constraints to the reefs in Belize via amino-acid racemization and U-Series ages of corals while Mosley et al. (2013) used speleothems to constrain late Pleistocene sea levels within the Mexican portions of the Yucatan Caribbean Sea.

## 3. Sea-Level Indicators

In the following discussion, I use "WALIS RSL ID" followed by a number to identify each of the RSL indicators discussed in the text that has been entered into the WALIS database. The number corresponds with the WALIS database identification numbers. Similarly, I use "WALIS U-Series ID" followed by a number to identify each of the RSLs identified from a single coral discussed in the text that has been entered into the WALIS online database. I also use "WALIS LUM ID", "WALIS ESR", or "WALIS AAR ID" followed by a number to reference optically stimulated luminescence ages, electron spin

resonance ages, or amino acid racemization ages, respectively, discussed in the text that have also been entered into the WALIS online database.

### 3.1 Gulf of Mexico

The LIG shoreline across most of the northern Gulf of Mexico formed what is locally known as the Ingleside Shoreline across Texas (Price, 1933) and the Gulfport Shoreline across Mississippi, Alabama, and the panhandle of Florida. Buried presumed

LIG deposits have been identified beneath the Mississippi River (e.g. Prairie Terrace; Fisk, 1944) but their sea-level significance is not well constrained (Otvos, 2005) and their elevations are likely significantly contaminated by sediment-loading induced subsidence.

### 3.1.1 Northwestern USA Gulf of Mexico (Texas)

Along the Texas Coast, the Ingleside Shoreline is composed of a +1 m to +8 m high, 3 to 16 km wide shore-parallel sand

deposit up to ~30 m thick (Wilkinson et a l., 1975; Paine, 1993; Simms et al., 2013). Cores through the feature contain many shell beds, intact oyster reefs, and bedding stratification indicative of an old beach system (Paine, 1993; Wilkinson et al., 1975). Its mixed transgressive and regressive nature suggests it was deposited near the maximum shoreline transgression of the LIG highstand in sea levels (Paine, 1993). It can be mapped intermittently from near Baffin Bay north to near the Texas/Louisiana border (Fig. 1), although the age and depositional origin for the most eastern portions of the shoreline mapped

by Fisher et al. (1973) are still debated (Otvos, 1997). In many locations, it surface still maintains the ridge and swale topography of old beach ridges (Fig. 2). Simms et al. (2013) obtained six OSL ages from four cores up to ~5 m in length from two portions of the Ingleside Shoreline. Three of the ages came from the core of the sand deposit, while three additional ages were obtained from the reworked top veneer of the feature. The three ages from the core of the feature were 119.0±7.0 ka (WALIS LUM_ID #145), 120.1±8.40 ka (WALIS LUM ID #119) and 127.9±8.70 ka (WALIS RSL ID #118) (Simms et al.,

2013). The younger ages ranged between 1.3 and 57.0 ka and all coincided with time periods of known heightened aeolian activity (Otvos, 2004; Simms et al., 2013).

The elevation of the Ingleside Shoreline varies across its expression in Texas. Paine (1993) provided one of the first attempts at a rigorous quantitative estimate of RSL change at the last interglacial based on the Ingleside Shoreline in Texas. In order to

quantify subsidence across the Gulf of Mexico at different time scales, Paine (1993) noted the maximum elevation of shell horizons in boring descriptions from a compilation of older geotechnical reports was 2-m above modern sea level (general definition) with the highest *in situ* oysters (*Crassostrea virginica*) at an elevation of 0.5 m. The indicative meaning of *Crassostrea virginica* is not well constrained but restoration efforts suggest they prefer water depths of 0.5 to 3 m or -1.75±1.25 m along the US Gulf Coast (Barnes et al., 2007; Pollack et al., 2012), although the lower bounds may simply reflect the scarcity

of water depths more than 3 m within Gulf of Mexico bays (Barnes et al., 2007). The Ingleside attains higher elevations but a portion of that elevation is late Pleistocene coastal dunes that covered the underlying beach deposits during progradation shortly after the time of beach deposition (Wilkinson et al., 1975). Paine (1993) subtracted the 2-m of elevation from a "global" 8-m sea-level highstand at the LIG to suggest a long-term subsidence rate of 0.05 mm/yr for the central Texas Coast. Although the indicative meaning of the shell horizon is poorly constrained, as it comes from Texas Highway Department and US Navy

boring descriptions (Paine, 1993), it likely represents deposition within a foreshore or barrier flat (landward side of a barrier island dominated by washover processes) environment, which do not extend to elevations of more than ~1 m above modern sea levels along the Texas Coast (Brown et al., 1976; Rodriguez et al., 2004; Simms et al., 2006). The shell deposits may have originated in deeper water. Although sandy deposits (e.g upper-shoreface sands) along the modern central Texas coast extend into water depths as great as 12 m (Rodriguez et al., 2001), they likely do not represent deposition within water depths any

greater than 2.5 m as water depths greater than that would place coeval Ingleside eolian deposits underwater (Wilkinson et al., 1975). Taken together this suggests an indicative meaning for the shell horizons of -0.75±1.75 m. Assuming the Ingleside eolian cap was originally eolian in origin and simply reworked by later dry phases of climate (Otvos, 2004; Simms et al., 2013) and the elevations reported by the Texas Highway Department and US Navy have an error of ±1 m, I assign a LIG sea-level range of +2.75±2.0 m for the shell horizons mentioned by Paine (1993)(WALIS RSL ID #915). For the *in situ* oysters, using

an indicative range of -1.75±1.25 m, suggests LIG sea levels greater than +2.25±1.6 m assuming an error of ±1 m for the elevations reported by the Texas Highway Department and the US Navy. We report the higher of these two sea-level estimates in the WALIS database but note that they are within error of one another.

     As it is difficult to assign a strict indicative meaning to the Ingleside deposits as presently described, Simms et al. (2013) took

a different approach to estimate paleo-RSL from the Ingleside by mapping the feature in an geographical information system (GIS) software package using soil survey maps and determining its elevation from the United States Geological Survey's (USGS) National Elevation Dataset (NED) digital elevation model (DEM). The NED has a published 95[th] percentile confidence level of 3.02 m (Gesch et al., 2014), but is more accurate (1.94 m) along regions surveyed by LiDAR, which includes most of the coastal counties (Gesch et al., 2014). Assuming the Ingleside was a LIG barrier island (Price, 1933; Paine,

1993) similar to the modern barrier islands of the Texas Gulf Coast, which is still a matter of discussion (Otvos, 2018, 2020), Simms et al. (2013) subtracted the average elevation of the closest modern equivalent barrier island from the elevation of each of the Ingleside "barrier island" segments of the Texas Coast. Assuming the preserved Ingleside deposits formed when RSLs reached their highest during the LIG, as many of them contained preserved beach ridges (Fig. 2b), and erosion led to little loss

in elevation of the feature, the resulting calculations lead to a range of RSL differences at the LIG across the Texas coast from a high of 7.2 m for the Vidor segment (WALIS RSL ID #778) to a low of 0.2 m for the Hoskins Island segment (WALIS RSL ID #774)(Fig. 3). However, the Vidor segment (Orange of Otvos, 1997) and the Hoskin Island segments may not represent barrier islands (Otvos, 1997) and have yet to be dated. If the Vidor segment represents a different age or depositional environment, then the highest non-contested LIG barrier island in East Texas would be the Fannett segment, which has well-preserved beach ridge features, with an average elevation of +5.8 m (WALIS RSL ID #777)(Otvos, 1997; Simms et al., 2013). Regardless, variability in the elevations of the LIG shorelines is more than 5 m, even after correcting for GIA-predicted variations across the Ingleside (Simms et al., 2013). This variability was interpreted by Simms et al. (2013) to represent differential subsidence across the Texas coastal plain with higher rates near the Brazos-Colorado delta complex. Simms et al. (2013) found a RSL difference of 2.8±4.9 m for the same segment of the Ingleside (Live Oak; WALIS RSL ID #772) that Paine (1993) found a shell horizon at an elevation of 2 m, which I assigned a RSL value of +2.75±2.0 m (Willow Creek; WALIS RSL ID #915), suggesting a reasonable result for the approach taken by Simms et al. (2013).

Simms et al. (2013) assigned an elevation error in their analysis of 1 standard deviation of the DEM pixel elevations of the Ingleside segment and modern islands. They also did not include an error term for the uncertainty in the DEM. Given the assumptions in their analysis of no erosion of the LIG features (either fluvial or eolian deflation), similar wave and wind climates and coastal sediment supplies at the LIG compared to today (and thus average barrier island elevations), this study takes a more conservative approach to the error by increasing the error for each of the estimates of Simms et al. (2013) to 2 standard deviations plus an additional error term of ±2.0 m to account for errors in the DEM. Another assumption in the analysis of Simms et al. (2013) and of the extension of that analysis for the eastern Gulf of Mexico below is that the modern and LIG equivalent barrier islands were both either transgressive or regressive barrier islands. One of the major geomorphic differences between transgressive and regressive barrier islands is their widths as generally transgressive barrier islands are thinner and lower than their regressive counterparts (Wallace et al., 2010; Otvos and Carter, 2013). In the absence of detailed sedimentological descriptions of the individual LIG segments, I use their widths as a reliability gauge of using a modern barrier island's average elevation as an analogue for their LIG equivalent. For the Texas Coast, the widths of the LIG barrier islands are similar to their modern barrier islands equivalents (Fig. 4), although some of the central Texas Ingleside segments are wider than their modern equivalents.

### 3.1.2 Northeastern USA Gulf of Mexico

The Gulfport Shoreline, in some locations also known as the Pamlico Shoreline, has a similar expression and elevation as the Ingleside of Texas (Otvos, 1972). It rises between +5 m and +9 m and can be traced from the Pearl River, Mississippi to near the Apalachicola delta of the Florida Panhandle (Otvos, 2005). It has been dated in four general locations using OSL. Along the Mississippi Coast, Otvos (2005) dated it in the city of Biloxi/Gulfport and Gautier. OSL ages there were 117.2±12.4 (WALIS LUM ID #146) and 124.0±10.8 ka (WALIS LUM ID #147), respectively. A third date in Mississippi near Bay Saint

Louis (English Lookout; WALIS LUM ID #200) returned an older age of 160±17 ka (Otvos, 2005). To the east at Morgan Peninsula near Mobile, Alabama, Blum et al. (2003) obtained two OSL ages of 137.8±34.4 ka (WALIS LUM ID #168) and 133.5±21.1 ka (WALIS LUM ID #168) for the locally termed "Pamlico Shoreline". Farther to the east in Florida the Gulfport Shoreline has been dated in two locations. In Gulf Breeze near Pensacola, Florida, Otvos (2005) obtained an age of 116.1±9.1 ka (WALIS LUM ID #148). Near Apalachicola, Florida, Burdette et al. (2012) obtained 17 OSL ages from the Gulfport Shoreline. The ages ranged from 108.7±8.2 ka to 138.7±11.1 ka (WALIS LUM ID #s 149-165) with an average age of 125.3±21.0 ka (error is 2 standard deviations of the ages). Four attempts at dating the Gulfport Shorelines along the eastern Gulf of Mexico using thermoluminescence were less conclusive with ages of ~38.8±3.7 ka to 103.0±10 ka (not included in WALIS) near the same location as the 117.2±12.4 ka OSL age from Biloxi/Gulfport (Otvos, 2005).

The original publications of Blum et al. (2003) and Otvos (2005) provide little information about the elevations of the Gulfport segments dated. However, Rodriguez and Meyer (2006) did collect GPR profiles through the beach ridges dated by Blum et al. (2003) on Morgan Peninsula, but do not show the lines collected over the LIG-aged deposits. However, they do report that the LIG (Sangamon in their original publication) beach ridges (*sensu* Otvos, 2020) were 4-5 m in height while the modern beach ridges were 2-3 m in height. I thus assign a modern analogue value of 2.5±1.0 m and a LIG elevation of 4.5±0.5 m for the LIG beach ridge elevation. This suggests a LIG RSL of +2.0±1.1 m (WALIS RSL ID #425). Burdette et al. (2012) used ground-penetrating radar (GPR) to identify the "transition point" between overwash fan deposits or aeolian sand sheets and the underlying foreshore deposits to be at an elevation of 3.75 m. Subtracting out the ~1 m elevation of the modern equivalent, they arrived at a RSL estimate for LIG at the Apalachicola Delta of +2.75 m (they originally report +2.5 m; WALIS RSL ID #411). No error was reported by Burdette et al. (2012) thus I assign an error of 1.0 m.

As many of the LIG sites along the northeastern Gulf of Mexico lack quantitative estimates of the elevation of RSL at the LIG, and assigning an indicative meaning to the deposits as currently described remains difficult, I followed the methods of Simms et al. (2013) to assign a RSL elevation for the LIG. This estimate was determined by subtracting the average elevations of the closest modern barrier islands from the average elevations of the six segments of the Gulfport Shoreline dated with the assumption that they too represented barrier islands (Figs. 5 and 6; Table 1). Mapping the margins of the LIG features along the northeastern Gulf of Mexico using soil surveys is not as straight forward as it is along the Texas coast due to the sandier nature of much of the northeastern Gulf of Mexico coastal plain and shelf (the Gulfport Shoreline is not bordered along its inland margins by a muddy unit as the Ingleside is in Texas). From this approach I obtained RSL estimates at the LIG for the English Lookout, Gulfport/Biloxi, Gautier, Morgan Peninsula, Gulf Breeze Florida, and Apalachicola sections of the Gulfport Shoreline as +3.6±4.6 m (WALIS RSL ID #891), +5.0±5.5 m (WALIS RSL ID #424), +3.1±3.8 m (WALIS RSL ID #892), +0.9±4.4 m (alternative elevation assignment for WALIS RSL ID #425), +5.0±7.4 m (WALIS RSL ID #893) and +2.9±4.2 m (alternative elevation assignment to WALIS RSL ID #411), respectively (Table 1). The value of +2.9±4.2 m for Apalachicola agrees well with the value obtained by Burnette et al. (2012) based on GPR of +2.75±1.0 m derived from the same segment of

the coast. In addition, the value of +0.9±4.4 m for Morgan Peninsula falls within error of the +2.0±1.1 m value obtained when simply using the published elevations for the modern and LIG beach ridges as reported by Rodriguez and Meyer (2006). In each case, I deferred to the elevations derived from the GPR and beach ridge survey, respectively, rather than the average elevations of the shoreline segments within the WALIS database.

Unlike many of the barrier islands in Texas, Morgan Peninsula displays a morphology suggestive of a complex and multi-stage evolution. Its complex architecture reveals an additional potential pitfall in using the average elevation of the LIG and modern barrier islands to determine the RSL difference at the LIG (Fig. 7). Morgan Peninsula experienced at least three phases of Holocene growth (Little Point Clear, Edith Hammock, and modern spit; Rodriguez and Meyer, 2006; Blum et al., 2003;

Fig. 7). Their elevations vary more among the three Holocene phases than with the LIG (Fig. 7), likely reflecting the variability in sea levels and wave climates at the time of their formation (e.g. Rodriguez and Meyer, 2006; Donnelly and Giosan, 2008). In addition, the barrier island geomorphology is not as apparent for the Gulfport Shoreline segments as it is for the Ingleside nor are the analogue barrier islands as similar in width as the Ingleside segments from Texas. Thus using the larger error bars for this analysis seems warranted.

### 3.1.3 Mexican Gulf of Mexico

A feature similar to the Ingleside appears to continue south along the Gulf of Mexico south of the USA/Mexico border to Soto la Marina, Tamaulipas (Price, 1958) and possibly farther south into Veracruz-Llave but has yet to be dated (Wilhelm and Ewing, 1972; Hernandez-Santana et al., 2016; Figs. 1 and 2). Near Soto la Marina these features are dotted with small ponds similar to the blow-out features common to the Ingleside across Texas (Price, 1933; Otvos, 2005; Simms et al., 2013).

However, their LIG age has not been verified and thus no data for these features has been input into the WALIS database. More work mapping and dating this potential LIG shoreline is warranted.

### 3.2 Yucatan Peninsula

Dated LIG beach ridges and reefs have been identified and studied across many locations of the Yucatan coastlines of Mexico and neighboring Belize (Fig. 8). Additional constraints on LIG sea levels from the Yucatan have been reported based on

speleothems within caves near the Mexican LIG beach ridges and coral reefs. These are the subject of a separate compilation within WALIS but are briefly discussed with reference to the other data reviewed in this study.

### 3.2.1 Mexico

A prominent set of LIG calcarenite beach ridges extends across much of the northeastern portion of the Yucatan (Szabo et al., 1978; Ward and Brady, 1979). The calcarenite beach ridge plain extends 150 km from Cancun to Xel Ha with a width of 0.5

to 4 km and thicknesses ranging from 3 to 10 m (Ward and Brady, 1979)(Fig. 8). The strandplain is underlain by a caliche developed over older Pleistocene coral-bearing limestones (Fig. 9). In addition, a few isolated *Diploria* and *Montastrea* corals

in growth position overlie the caliche crust but are covered by the overlying calcarenite beach deposts (Ward and Brady, 1979; Szabo et al., 1978).  Szabo et al. (1978) obtained 5 U-series ages on corals reworked into the calcarenite beach ridge deposits as well as the underlying isolated *in situ* corals.  Three of the corals reworked into the overlying calcarenite beach deposits dated to 121±6 ka, 123±6 ka, and 120±6 ka (WALIS RSL ID #438).  *In situ* or only lightly abraded corals of *Montastrea* sp. found ontop of the caliche crust dated to 123±6 ka (WALIS RSL ID #439) and 125±15 ka (WALIS RSL ID #440; Szabo et al., 1978).  In addition to the corals within and underlying the calcarenite beach ridges between Cancun and Xel Ha, Szabo et al. (1978) also obtained an U-series date from a reef coral on the nearby island of Cozumel that returned an age of 121±6 ka (WALIS RSL ID #441).

Although the calcarenite beach ridges reached elevations of 10 m (Szabo et al., 1978), they are capped by an eolianite facies (Fig. 9).  The base of the calcareous beach facies with cross-bedding lies at elevations of +3.5 m to +6.5 m above present sea level (general definition; Szabo et al., 1978).  A more detailed stratigraphic description of the deposits by Ward and Brady (1979) suggests the boundary between the upper shoreface and foreshore/backshore is found at ~+4.8 m across the calcarenite strandplain.  Based on the difference in elevation between that LIG contact and the modern upper shoreface/forshore contact (contact elevation not given), Ward and Brady (1979) argue that RSL during the LIG was between +5 m and +6 m. This assessment is also based on the assumption that the beach ridges and associated upper shoreface and foreshore deposits represent the highstand rather than a later regressive phase at a potentially lower stage of sea level.  I attempt to provide a more quantitative approach to estimating LIG RSLs based on the data of Ward and Brady (1979) by assuming the foreshore/upper shoreface contact represents the "breaking depth" of the waves.  Using the IMCalc application of Lorscheid and Rovere (2019) for this region yields a breaking depth of -1.4±1.0 m.  Combined with an elevation of the upper shoreface/foreshore contact at +4.8 m, with an assumed ±1.0 m of error to account for the uncertainties associated with how the elevation was measured, what datum was used, and the details of the modern analogue, yields a LIG RSL of +6.2±1.4 m (WALIS RSL ID #438).  The two lower elevation *Montastrea* corals, which inhabit water depths of -1.1 m to -17 m (Lightley et al., 1982; Hibbert et al., 2016) at +2 m as well as a third on the nearby island of Cozmel also at +2 m provide limiting constraints on LIG RSL that are consistent with the +6.2±1.4 m RSL assignment.

Ward and Brady (1979) also noted an extensive tract of Pleistocene coral reefs seaward of the cacalrenite beach ridges.  The reef tract contains *in situ* corals of *Montastrea annularis*, *Acropora cervicornis*, and rarely *Porites porites* that make an intact coral reef structure (Jordán-Dahlgren, 1997).  The *A. cervicornis* reef unit is found at elevations as high as +4.5 m, with overlying storm deposits with reworked *Acopora palmata* corals found at elevations as high as +5.5 m (Jordán-Dahlgren, 1997).  In some locations this complex is overlain by the distal extension of the calcarenite grainstones.  Blanchon et al. (2009) and Blanchon (2010) conducted an extensive description and dating of this reef tract in a theme park at Xcaret (Fig. 8).  They were able to document two levels of well-preserved *in situ* reef structures and reconstruct a cross-section through the reefs, which allowed them to confidently identify the reef flat and reef crest of the Pleistocene reefs (Fig. 10).  They obtained

33 U-series ages on corals from the two different reef tracts. The U-series ages ranged from 107.7±1.0 ka to 158.9±1.9 ka, with two additional outliers of 179±2.4 ka and 567.3±139.1 ka. Removing those two outlies gives an average age of the reef tract of 126.7±27 ka (error is 2 standard deviations of the ages). Blanchon et al. (2009) screened the 33 samples removing those whose $^{238}U$, $^{232}Th$, and $\delta^{234}U(T)$ values were not the same as modern seawater. This filtering left 2 samples from the lower reef and 5 samples from the upper reef from well-preserved corals. The reliable ages from the lower reef tract returned ages of 132.6±1.6 ka and 134.3±1.4 ka (WALIS RSL ID #446). Of the five samples from the upper reef, two were from reworked clasts of *A. palmata*. The three remaining ages from the upper tract were 117±1.1 ka, 117.7±1.1 ka, and 119.5±1.1 ka (WALIS RSL ID #445); two of these ages were from *in situ Acropora palmata* colonies (Blanchon et al., 2009). Both sets of ages were recalculated by Hibbert et al. (2016). This new recalculation based on updated decay constants (e.g. Cheng et al., 2013) and spike corrections (Hibbert et al., 2016) suggests ages of 131.6±0.9 ka and 135.1±0.9 ka for the lower reef tract (WALIS RSL ID #446) and 117.5±0.5 ka, 118.2±1.5 ka, and 121.3±0.6 ka for the upper reef tract (WALIS RSL ID #445).

The well-documented framework of the ancient reef systems allowed for the identification of the different segments of the LIG reef at Xcaret (Jordán-Dahlgren, 1997; Blanchon et al., 2009; Blanchon, 2010)(Fig. 10). The LIG reef crest currently lies at an elevation up to +5.8 m (the highest dated *in situ* colonies were obtained from +4.9 m, Blanchon, 2010) for the upper reef tract and +3 m for the lower reef tract (Blanchon et al., 2009)(Fig. 10). Both reef crest deposits contain in place *A. palmata* colonies (Blanchon, 2010). Stated uncertainties for the elevations from Blanchon et al. (2009) are ±0.15 m, one half the tidal range of 0.3 m. Based on their stratigraphic relationships and differences in age, Blanchon et al. (2009) interpret the upper reef tract to represent a later highstand within the LIG and the lower reef tract to represent an earlier and lower phase of sea level during the LIG. They interpret the two different elevations of sea levels at the LIG as a result of a rapid sea-level rise at the LIG. Based on a comparison with modern equivalents, Blanchon (2010) argues that the reef crests were deposited in intertidal conditions and thus represent a LIG RSL highstand of +6 m. A study by Cubit et al. (1985) that included a detailed water-level survey across a Caribbean reef flat in nearby Panama found that the reef flat within a similar tidal range developed 6 cm below mean lower low water. Thus, based on half the tidal range of 30 cm and an elevation of +5.5 m for the fossilized LIG reef flat, we assign an LIG RSL of +5.7±0.2 m. That assignment places the coeval reef crest 10 cm above mean sea level. As it appears that the reef flat of the lower reef tract reaches an elevation of +2.5 m, the same approach applied to the lower reef tract suggests an early LIG RSL of +2.7±0.2 m. The error terms of ±0.2 are the root sum of the squares of one half of the indicative range (0.15 m) and the measurement error, ±0.15 m.

Inland and only a few 10's of km to the south of these LIG calcareous beach ridges and coral reefs, Moseley et al. (2013) surveyed and dated 10 subaerially formed speleothems from the cave networks south of Xel Ha in Quntana Roo, Mexico (Fig. 8). A total of 50 U-series ages were obtained from these speleothems. The ages ranged from 59.3±0.4 ka to 117.7±1.4 ka (Moseley et al., 2013). The speleothems were obtained from elevations of between +1.5 and -15.1 m relative to modern sea level (general definition). Their elevations were determined based on a digital depth gauge relative to the modern water table

with stated accuracies of ±0.1 m with a maximum salinity-driven water density conversion uncertainty of 2% (Moseley et al., 2013). The growth of the speleothems provide only limiting information on past sea levels and largely constrain the maximum elevations of sea levels during the late LIG, marine isotope stage (MIS) 5c, and MIS5a (Moseley et al., 2013). Nevertheless, they suggest sea levels during the LIG dropped below -4.9 m by 117.7±1.4 ka assuming a subsidence rate of 0.001 m/ka (Emery and Uchupi, 1972; Moseley et al., 2013)(Note: the subsidence correction only accounts for 0.1 m over the 117.7 ka).

**3.2.2 Belize**

Within Belize, LIG corals have been found onshore at Ambergris Cay as well as within drill core beneath the Turneffe Islands, Lighthouse Reef, and Glovers Reef (Gischler and Hudson, 1998; Gischler and Lomando, 1999; Gischler et al., 2000)(Fig. 11). These corals have been dated using U-series ages by Gischler et al. (2000) and Mazzullo (2006). In addition, Mazzullo (2006) obtained two additional amino acid racemization ages from the corals.

U-series ages obtained from Reef Point at Ambergris Cay dated to 128.28±1.33 ka (WALIS U-Series ID #1693) by Gischler et al. (2000) and 135.8±0.9 ka (WALIS RSL ID #448) by Mazzullo (2006)(Fig. 11). These ages were obtained from *A. palmata* and *M. annularis* at elevations of 0.3 and 0.5 m, respectively. Mazzullo (2006) obtained a second U-series age from another *M. annularis* coral dredged from 2.3 m depth that dated to 165.5±1.1 ka, but was deemed unreliable given its high Th content.

Mazzullo (2006) describe the facies found at Reef Point on Ambergris Cay as a reef flat. With a tidal range of 0.3 m (Gischler and Lomando, 1999) and treating the reef flat as forming at mean lower low water (Cubit et al., 1986), places LIG RSLs at +0.45±1.0 (WALIS U-Series ID #1693) and +0.65±1.0 (WALIS RSL ID #448). The error was derived from the root square of the sum of half the indicative meaning (0.15 m) and 1.0 m to account for uncertainties in the elevation measurement datum and modern analogues. Two specimens of the gastropod *Strombus gigas* gave AAR ages equivalent to the LIG from an

elevation of +1.2 m approximately 7 km southwest of Reef Point (WALIS AAR ID #s 129,130; Mazzullo, 2006). The gastropod inhabits very shallow waters but can be found in water depths as great as 60 m (Randall, 1964), and thus only confirms the RSL elevation limits placed by the corals but does support the age assignment of the reefs.

The other 6 U-series ages of Gischler et al. (2000) were obtained from cores taken on Glovers, Lighthouse, and Turneffe reefs

(Fig. 11). Neither of the two samples obtained from cores on Glovers reef was considered reliable by Gischler et al. (2000) as they both appeared too old. One dates to 280.3±3.0 ka (WALIS U-Series ID #1695), while the other dates to 138.0±0.8 ka (WALIS U-Series ID #1694). Although the facies of these limestones are conclusively barrier reef in origin, they are not as well described as those from Ambergris Cay and the indicative meaning of the deposits is not clear. Therefore, I initially turned to the general habitat zones of the coral species dated. Both ages were obtained from specimens of the coral *M. annularis*

that when using the depth ranges of Hibbert et al. (2016) would result in an RSL of 0.7+8.6/-7.3 m (WALIS U-Series ID #1695) and 1.7+8.6/-7.3 m (WALIS U-Series ID #1694). However, Gischler et al. (2000) mentions that the facies the coral were obtained from also includes specimens of the coral *A. palmata*, which would suggest lower sea levels on the order of -

7.5 +1.2/-7.9 m (WALIS U-Series ID #1695) and -6.5 +1.2/-7.9 m (WALIS U-Series ID #1694), respectively. The two samples from Lighthouse reef were both considered reliable by Gischler et al. (2000) based on their ages of 125.0±0.4 ka (reported as 124.99±0.355 ka; WALIS U-Series ID #1696) and 129.9±0.5 ka (WALIS U-Series ID #1697). The ages were obtained from specimens of *A. cervicornis* and *A. palmata*, respectively, which using the zonations of Hibbert et al. (2016) suggests LIG RSLs of -4.9 +4.1/-11.8 m (WALIS U-Series ID #1696) and -6.5 +1.2/-7.9 m (WALIS U-Series ID #1697), respectively. If we use the depth limits of the accompanying *A. palmata* for the former of these two samples, it might suggest RSLs as deep as -8.0+1.2/-7.9 m (WALIS U-Series ID #1996). The last two ages obtained from the same section of the core from Turneffe reef returned ages of 142.0±0.5 ka (WALIS U-Series ID #1698) and 145.3±0.5 ka (WALIS U-Series ID #1699). Gischler et al. (2000) also interpreted these to represent erroneously old ages. These two samples were obtained from specimens of *A. palmata* and *M. annularis*, respectively, and would suggest RSLs of-2.5 +1.2/-7.9 m (WALIS U-Series ID #1698) and 5.7 +8.6/-7.3 m (WALIS U-Series ID #1699), respectively. Concerning the latter of these two ages, similar to the samples from Glovers reef, the *M. annularis* likely represents an environment on the shallower end of the spectrum considering its association with specimens of *A. palmata* and thus may represent an RSL as shallow as -2.5 +1.2/-7.9 m (WALIS U-Series ID #1699).

The elevation of the top of the Pleistocene section beneath the reefs are much lower in Belize than those farther north along the Yucatan Peninsula near Cancun, Mexico (Gischler et al., 2000). In addition, the top of the Pleistocene appears to deepen to the south and east (Gischler et al., 2000). Gischler et al. (2000) attribute this to tectonic subsidence as the margin trails off into the adjacent Cayman Trough. This interpretation is supported by evidence of neotectonic activity found within Holocene coastal successions (McClowsky and Liu, 2013) and deeper (Lara, 1993) but the accuracy of the ages of the corals from Gischler et al. (2000) is still a matter of discussion (MacIntyre and Toscano, 2004). MacIntyre and Toscano (2004) suggest the possibility that the ages are erroneously too old given their relatively low aragonite percentages and elevated $^{234}$U/$^{238}$U ratios and the lower elevations actually reflect deposition during later substages of MIS5 (e.g. MIS5a, MIS5c, etc.). Additionally, the stratigraphic section described by Mazzullo (2006) with a U-series age consistent with the LIG as well as two amino-acid racemization ages at a similar elevation is capped by an unconformity. It remains to be determined if the LIG reefs grew higher and were subsequently eroded during the LGM, if these reefs represent transgressive or later regressive reefs formed before or after the LIG highstand, or whether the elevation of the corals at Ambergris Cay represent the LIG highstand in Belize with the subsequent lower elevations to the south and east along the Belize margin a reflection of neotectonic activity.

**3.3 Honduras**

Only a handful of possible LIG deposits have been located in Honduras. Cox et al. (2008) obtained an ESR age (WALIS ESR ID#102) on an uplifted fossil reef on the western tip of Roatan Island (WALIS RSL ID #450; Fig. 11). The poor preservation of the reef made it difficult to ascertain the elevation of RSL at the time of deposition and the corals are of unknown species. Late Pleistocene limestones with *in situ* specimens of *Montastrea sp*. and *Acropora cervicornis* have also been reported from

the Swan Islands (Ivey et al., 1980; Fig. 11) but have yet to be dated. They reach elevations up to 14 m above modern sea level. Both regions are likely heavily influenced by tectonic activity due to their development and growth across uplifted tectonic blocks along the Montagu/Swan Islands fault system (Cox et al., 2008). The mainland coast of Honduras is a well-developed siliciclastic coastline with prevalent presumably Holocene beach ridges but no LIG shorelines have been mapped across it to date.

## 4. Elevation Details

### 4.1 Datums

With the exception of the new work in this study and the works of Burdette et al. (2012) and Simms et al. (2013), little detail is given as to the datums of the LIG shoreline elevations. This study, Burdette et al. (2012), and Simms et al. (2013) utilize a sea-level datum of North American Vertical Datum of 1988 (NAVD88; https://www.ngs.noaa.gov/datums/vertical/north-american-vertical-datum-1988.shtml, last accessed June 22, 2020), which locally can vary from mean sea level by a meter or more (Kinsman and Youngman, 2018). However, in this study and that of Simms et al. (2013) both the LIG and the modern equivalent barrier islands were measure with respect to NAVD88 "0" and thus the differences between MSL and NAVD88 are cancelled out. Within the region of Burdette et al. (2012), the closest National Oceanic and Atmospheric Administration (NOAA) tide gauge station (8728690; https://tidesandcurrents.noaa.gov/datums.html?id=8728690, last accessed June 22, 2020) suggests a difference between MSL and NAVD88 of 0.15 m.

### 4.2 Elevation measurements

The rest of the studies defined mean sea-level according to the generic definition and provided little detail as to how the elevations were physically measured. Moseley et al. (2013) used a depth gauge while Burdette et al. (2012) and Simms et al. (2013) used high-resolution LIDAR with accuracies of 0.25 cm. However, within the entire region, the tidal range is less than 1 m, with some areas (e.g. the Yucatan) experiencing a tidal range of less than 0.15 m (Blanchon et al., 2009) and thus any errors associated with estimating the mean tide level are likely minimal and less than 1 m.

## 5. Related sea-level topics

### 5.1 Subsidence

With the exception of the Honduran coast and possibly the eastern Gulf of Mexico (Otvos, 1981), the currently-dated LIG sites across the northwestern Gulf of Mexico and northwestern Caribbean are all subject to subsidence rather than tectonic uplift. Within the northwestern Gulf of Mexico subsidence appears to increase basinward (Simms et al., 2013) and along the Belize Coast it appears to increase to the south and east (Gischler et al., 2000). However, constraining the magnitude of subsidence

independent of the LIG elevations has remained problematic as most studies use the elevation of the LIG shoreline to determine subsidence (e.g. Paine, 1993; Gischler et al., 2000; Simms et al., 2013). Studies independent of the LIG shoreline elevation are needed to determine subsidence rates and hence correct LIG sea levels from its influence. GPS surveys provide some hope, but issues related to anthropogenic groundwater and hydrocarbon extraction are not always easy to correct for and likely dominate the subsidence signal at GPS timescales. Groundwater and hydrocarbon extraction are particularly relevant across the northern Gulf of Mexico (Paine, 1993; White and Morton, 1997; Morton et al., 2006; Chan and Zoback, 2007; Qu et al., 2015)

## 5.2 LIG sea-level fluctuations

With the exception of the study by Blanchon et al. (2009) most of the studies of the LIG shoreline across the Gulf of Mexico and western Caribbean have been too coarse to test for fluctuations in LIG sea levels. Most ages have only been precise enough to establish an LIG age and not necessarily when during the LIG the feature was deposited. Neither have the deposits lent themselves to reconstructing fine-scale fluctuations in sea levels during the LIG, particularly within the siliciclastic shorelines of the northern Gulf of Mexico. The carbonate systems of the Yucatan Peninsula may provide more opportunities for testing for sea-level fluctuations during the LIG. The exception is the work of Blanchon et al. (2009). They found two distinct reef tracts that they argue represent an earlier, lower phase of LIG sea levels at +3 m and a later higher phase of LIG sea levels at +6 m, separated by a rapid increase in LIG sea levels (Blanchon et al., 2009)(Fig. 10).

## 5.3 Earlier Highstands

Shorelines and other coastal features from highstands in sea levels prior to the LIG have been reported from the northern Gulf of Mexico but have yet to be dated (Winker and Howard, 1977; Donoghue and Tanner, 1992). The most studied and best preserved are those within the panhandle of Florida near the Apalachicola delta, where Winker and Howard (1977) and Donoghue and Tanner (1992) describe two older terrace and shoreline sets – the Gadsen and Wakulla sequences, the former of which may correspond to multiple highstands (Winker and Howard, 1977). However, some discussion has arisen as to their origin with some studies attributing these features to non-marine sources (Otvos, 1995) as very little detailed sedimentology has been conducted on the features to show their marine origins. In addition to the purported marine shorelines, the mapping of alluvial terraces suggests a progradational nature to much of the coastline with earlier phases of transgression and regression leading to the development of multiple periods of coastal plain aggradation (Otvos, 2005). However, the alluvial terraces have only been preliminarily dated (e.g. Otvos, 2005) and more work is required to nail down their ages and relationship to former sea levels.

Older Pleistocene reefal units are present across the Yucatan Peninsula (e.g. Ward and Brady, 1979; Ferro et al., 1999; Gischler et al., 2010) but have not been well dated nor been used to constrain the elevations of pre-LIG highstands. Speleothems that

may help constrain older sea levels dating as far back as MIS11 have been identified within Quintana Roo (Steidle et al., 2020). Those results have yet to be published outside of meeting abstracts, but are likely forthcoming.

## 5.4 Holocene sea-level indicators

Middle-to-late Holocene sea levels are well constrained in the region with several site-specific reconstructions as well as compilations available for the northern Gulf of Mexico (Tornqvist et al., 2004; Simms et al., 2007; Milliken et al., 2008; Livsey

and Simms, 2013) as well as the Caribbean (Toscano and Macintyre, 2003; Gischler and Hudson, 2004; Khan et al., 2017). The records become sparser for the early Holocene and late glacial periods. One discussion that has repeatedly resurfaced within the northern Gulf of Mexico is the possibility of a mid-Holocene highstand (e.g. Tanner et al., 1989; Blum et al., 2002) but currently appears to have fallen out of favor (Otvos, 2001; Simms et al., 2009).

## 5.5 Uncertainty and data quality

The amount of uncertainty in the age and elevation of the LIG sea-level indicators varies by location. The shoreline along the northern Gulf of Mexico is likely LIG in age but very few of the existing ages have the accuracy or precision to determine when within the generally accepted 115–129 ka time period it formed. The average error of the 24 OSL measurements thought to have been derived solely from LIG deposits is 10.4 ka, far too large to determine when within the LIG the feature(s) formed. Because few of the studies on the LIG shoreline to date have included detailed facies descriptions of the shoreline deposits,

the elevations are probably accurate to within 2-3 m of the former highstand elevation and likely larger for the DEM-derived elevations given the assumptions related to analogous LIG and modern barrier islands. This latter assumption includes uncertainties related to post-depositional erosion, similarities in wave climate and sediment supply, differences in transgressive versus regressive architectures, the interpretation of the LIG shorelines as paleo barrier islands, and specific timing of deposition with respect to the true highstand during the LIG. In addition, the lack of estimates of subsidence independent of

the LIG elevation at each site also contributions to the uncertainty of LIG RSLs along the Gulf of Mexico. This uncertainty due to subsidence is likely on the order of <5 m (Paine, 1993; Simms et al., 2013), but these estimates are in need of analyses independent of the LIG shoreline elevations.

The data from the northeastern Yucatan Peninsula probably provides the best estimates of RSL during the LIG for the region

surveyed in this study. The analysis of Blanchon et al. (2009) includes the most detailed facies analysis of coral reef deposits within the region leaving LIG RSL elevation estimates to within <1 m. In addition, their screened U-Th ages appear to be able to distinguish early from late LIG times. The earlier study of the Mexican Yucatan Peninsula by Szabo et al. (1978) are probably as accurate and precise as the estimates from the Gulf of Mexico with U-Th age error bars on the order of 6 ka and elevations probably good to the order of 2-3 m. For Belize, the U-Th ages of Gischler et al. (2000) have reported errors of less

than 1 ka but 4 of the 7 fall outside the generally accepted age range of the LIG. They may suffer from the effects of diagenesis (MacIntyre and Toscano, 2004). In addition, without definitive indicative meaning for the facies in which the corals dated

were obtained, I was left to rely on the stated depth ranges of Hibbert et al. (2016). These constraints are limited to consideration of the corals as individual species and not necessarily the suite of species present within the host facies. In addition, with such large age errors, the corals could represent reefs from the transgression leading up to the LIG or the regression that shortly followed. In addition, along with the neighboring constraint from Honduras, the sites from Belize are likely contaminated by vertical tectonic motion.

## 6. Concluding Remarks

The LIG shoreline is well expressed over portions of the northern and western Gulf of Mexico and the eastern Yucatan Peninsula. The Gulf of Mexico shorelines are largely the remnant of sandy shorelines and barrier islands while those of the Yucatan peninsula are both coral reefs and calcarenite beaches. The elevation of these features suggests local LIG sea levels were between +2 and +6 m across the region. However, these estimates from the northern Gulf of Mexico are not based on detailed sedimentary facies analysis and include several assumptions relating the similarity between modern and LIG depositional environments. In addition, they may be contaminated by subsidence, particularly within the Gulf of Mexico and potentially Belize. Although not well studied, tectonic uplift likely contaminants the elevation of the LIG shorelines within Honduras and its offshore islands. The best estimates of LIG sea levels within the region are probably those derived from the corals of the northeastern Yucatan Peninsula (e.g. Blanchon et al., 2009), which appears to be the most stable area within the region. Much work remains to be done in dating and mapping the LIG shoreline within northeastern Mexico across the border from the USA as well as within Honduras.

## 7. Data availability

The Gulf of Mexico and northwestern Caribbean Sea Last Interglacial sea-level database is available open access, and updated as necessary, at the following link: https://doi.org/10.5281/zenodo.4556163 (Simms, 2020). The files at this link were exported from the WALIS database interface on August 26, 2020. Description of each field in the database is contained at this link: https://doi.org/10.5281/zenodo.3961543 (Rovere et al., 2020) and is accessible (and searchable) here: https://walis-help.readthedocs.io/en/latest/. More information on the World Atlas of Last Interglacial Shorelines can be found here: https://warmcoasts.eu/world-atlas.html. If you use our database, we encourage you to cite the original sources alongside with this article.

## Author contribution

AS read the papers, compiled the data, conducted the ArcGIS analysis, and wrote the manuscript.

## Acknowledgements

I would like to thank Ian Baxter for taking an early look at the LIG elevations across the northeastern Gulf of Mexico and Paul Blanchon for sharing some of his original figures from Xcaret. John Anderson, Tony Rodriguez, and Regina DeWitt are thanked for their discussions regarding the LIG shorelines across the northwestern Gulf of Mexico. Paul Blanchon and Michael O'Leary provided insightful official reviews while Barbara Mauz also provided helpful comments. The data used in this study were compiled in WALIS, a sea-level database interface developed by the ERC Starting Grant "WARMCOASTS" (ERC-StG-

802414), in collaboration with PALSEA (PAGES / INQUA) working group. The database structure was designed by A. Rovere, D. Ryan, T. Lorscheid, A. Dutton, P. Chutcharavan, D. Brill, N. Jankowski, D. Mueller, M. Bartz, E. Gowan and K. Cohen.

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

**Figures:**

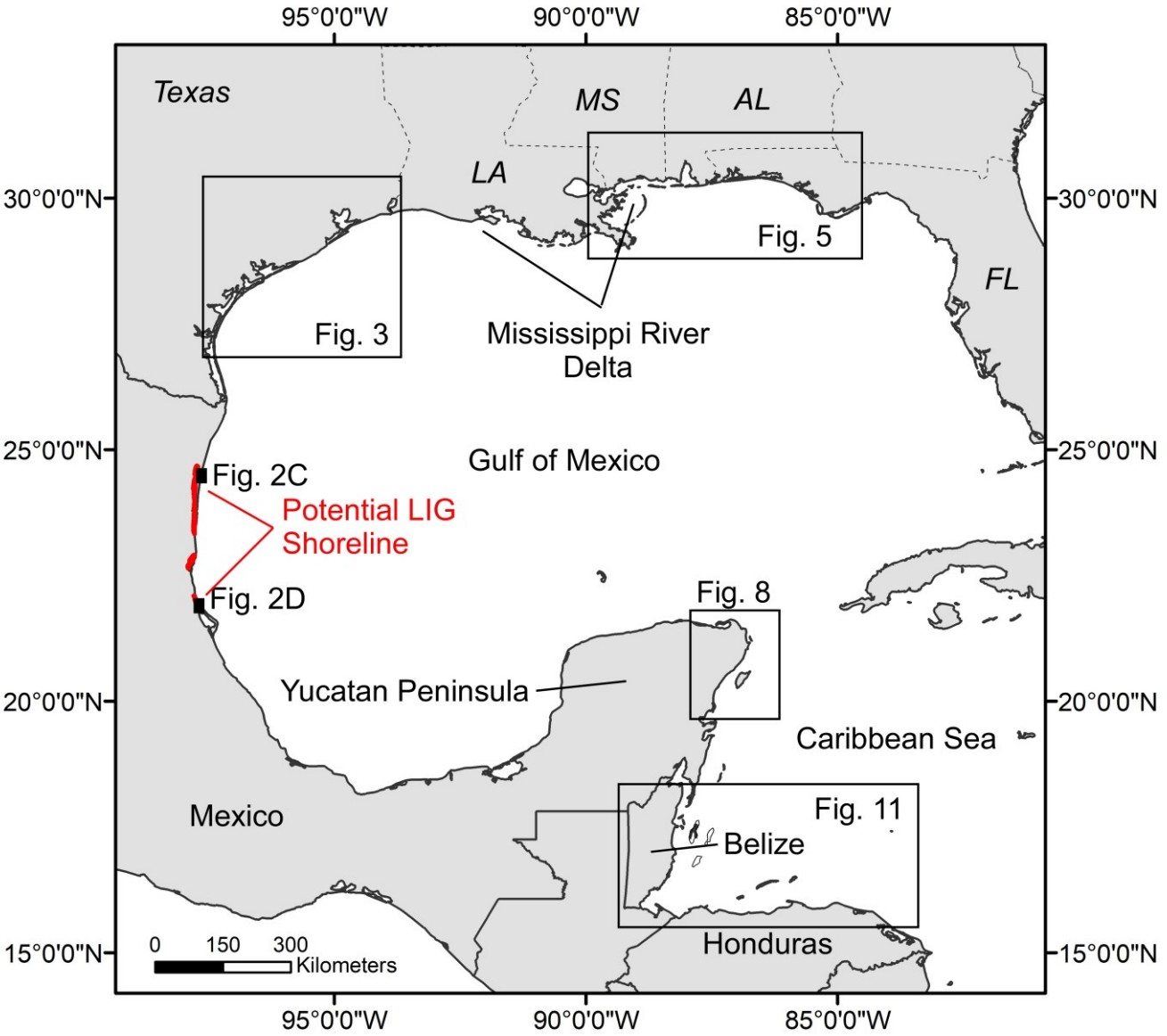

**Figure 1.** Map illustrating the location of the figures and locations mentioned in the text. LA = Louisiana, MS = Mississippi, AL = Alabama, and FL = Florida.

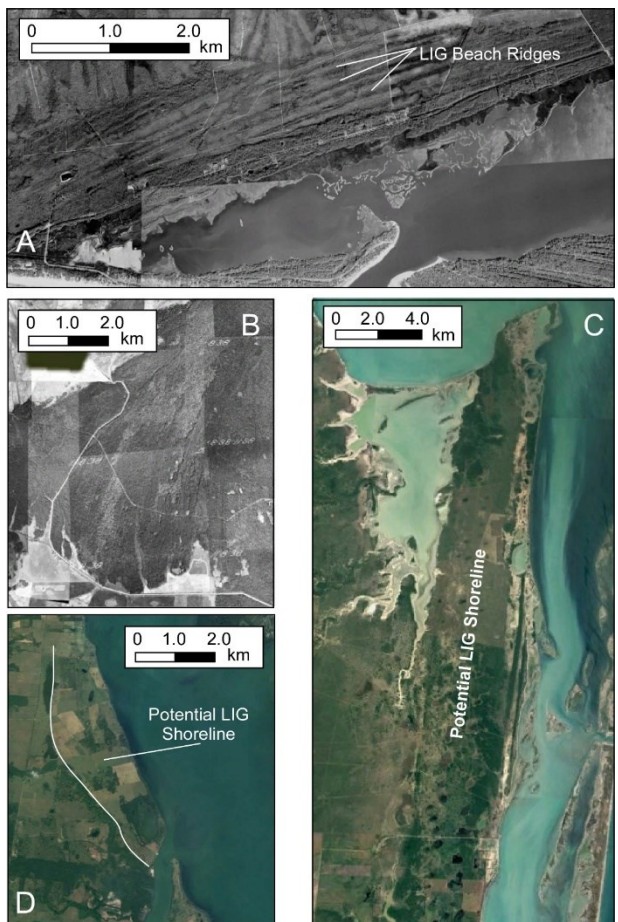

**Figure 2.** Aerial Photographs illustrating well preserved beach ridges on the LIG shorelines within A.) the Gulfport Shoreline near the Apalachicola delta (Florida)(USGS, 1993) and B.) the Fannett (Texas) segment of the Ingleside Shoreline (Texas General Land Office, 1938) in addition to the potential LIG shoreline equivalents to the Ingleside and Gulfport Shorelines northeast of Soto la Marina, Tamaulipas, Mexico (C.) and south of Tampico in Veracruz, Mexico (D.). Images from ©GoogleEarth.

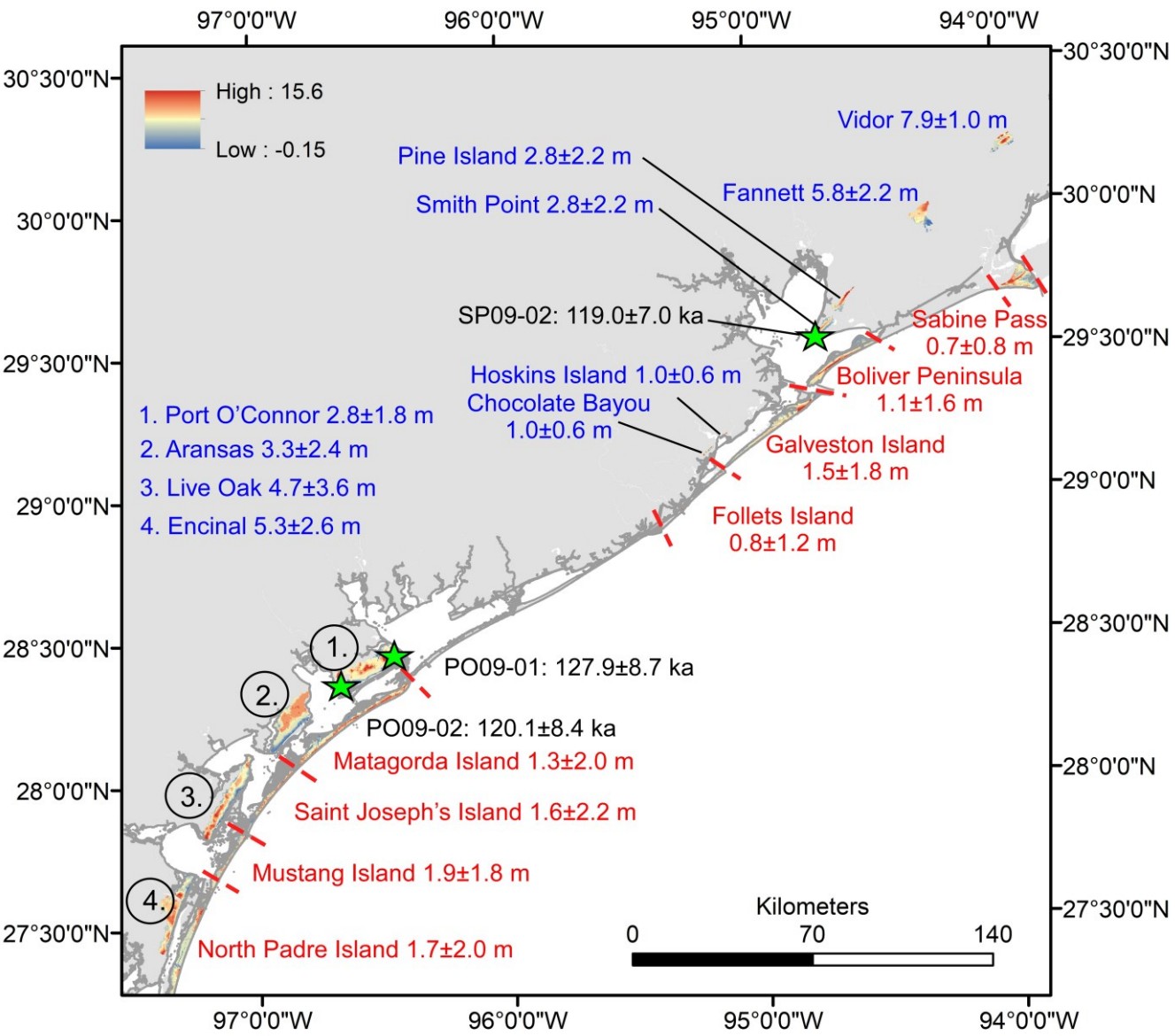

**Figure 3.** Map of the Texas coast of the northwestern Gulf of Mexico showing the locations of the LIG Ingleside shoreline segments (blue text) and modern barrier islands (red text) discussed in the text with their average elevations. Also shown as green stars and black text are the optically stimulated luminescence ages obtained from the Ingleside shoreline.

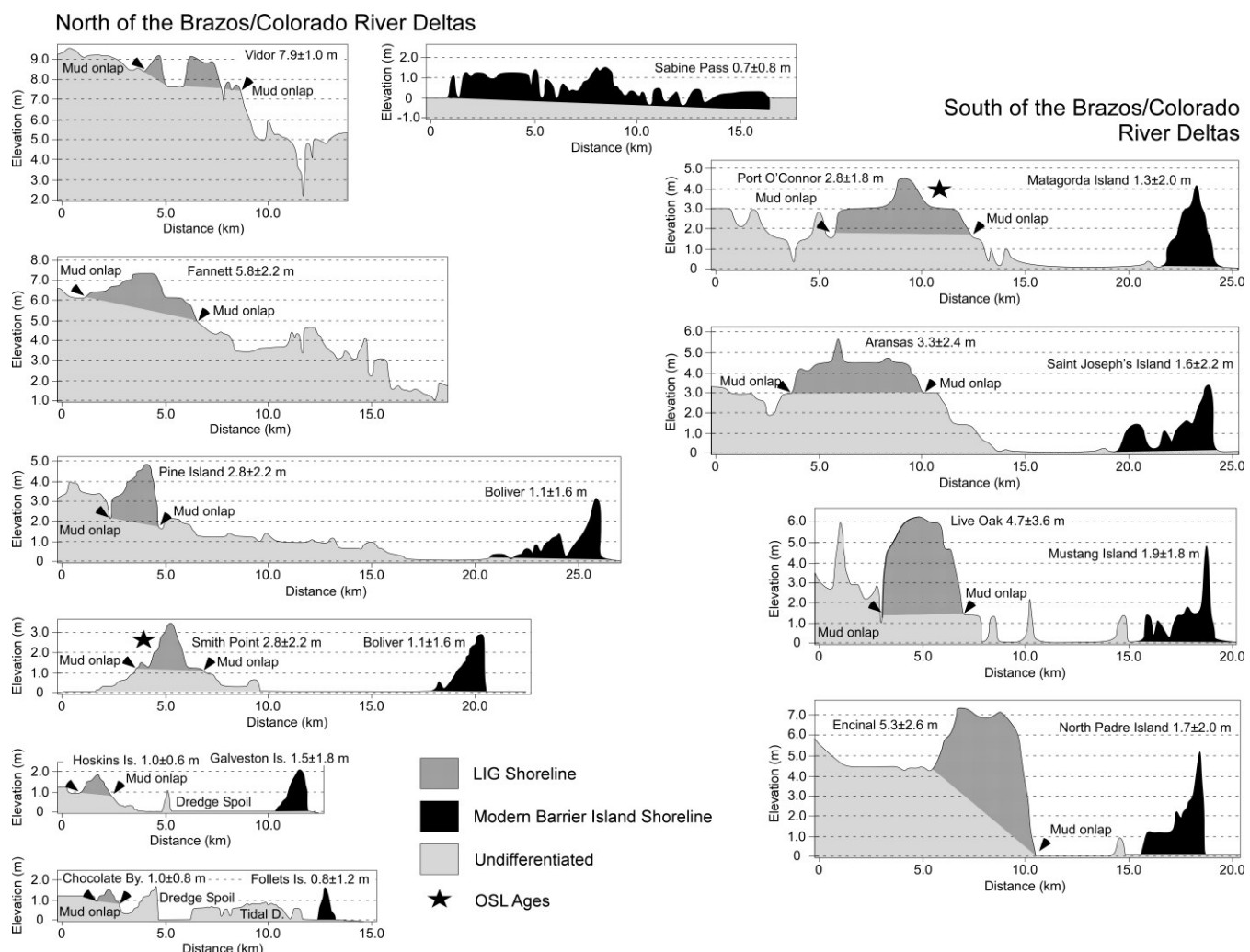


**Figure 4.** Topographic profiles through selected portions of the Texas coast illustrating the elevation and width differences between the Ingleside and modern barrier islands. See Figure 3 for general locations.

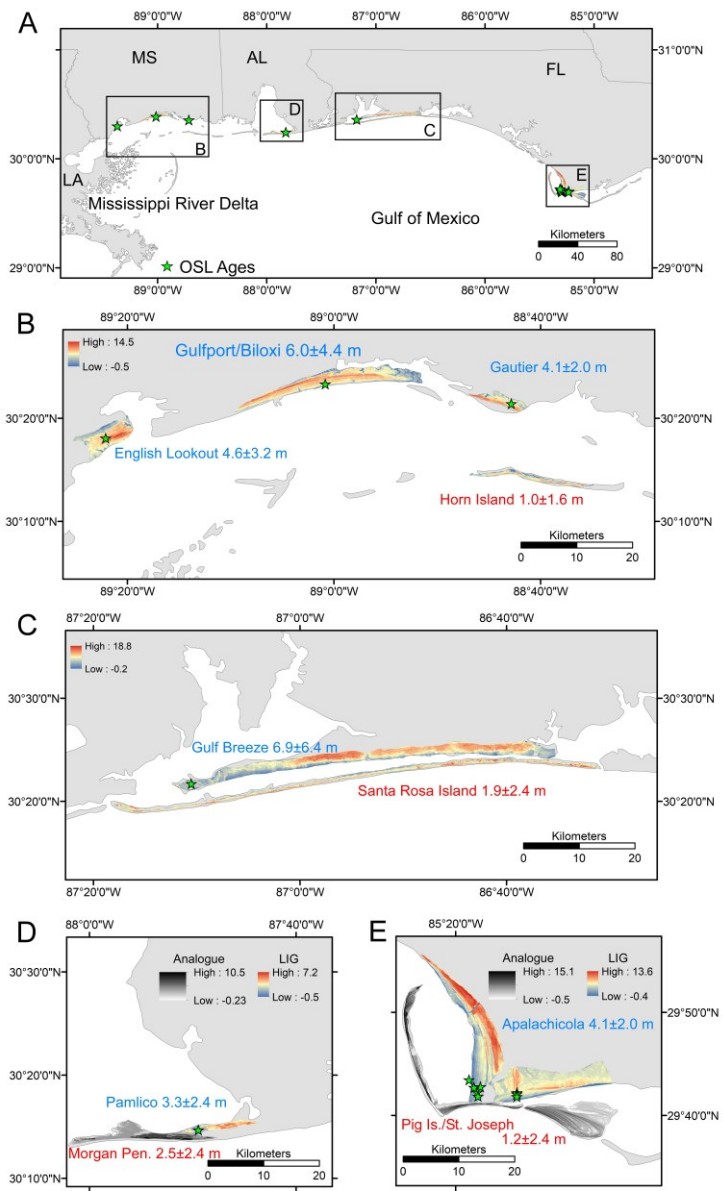

**Figure 5.** A.) Map of the LIG Gulfport shoreline across the northeastern Gulf of Mexico. Digital elevation models (DEM) of the LIG Gulfport shoreline segments dated along the Mississippi (B), western Florida Panhandle (C), Alabama (D), and Apalachicola Delta (E) regions. Stars mark the locations of optically stimulated luminescence ages. The LIG shoreline segment names and average elevations are given in blue text while the modern barrier island names and average elevations are given in red text.

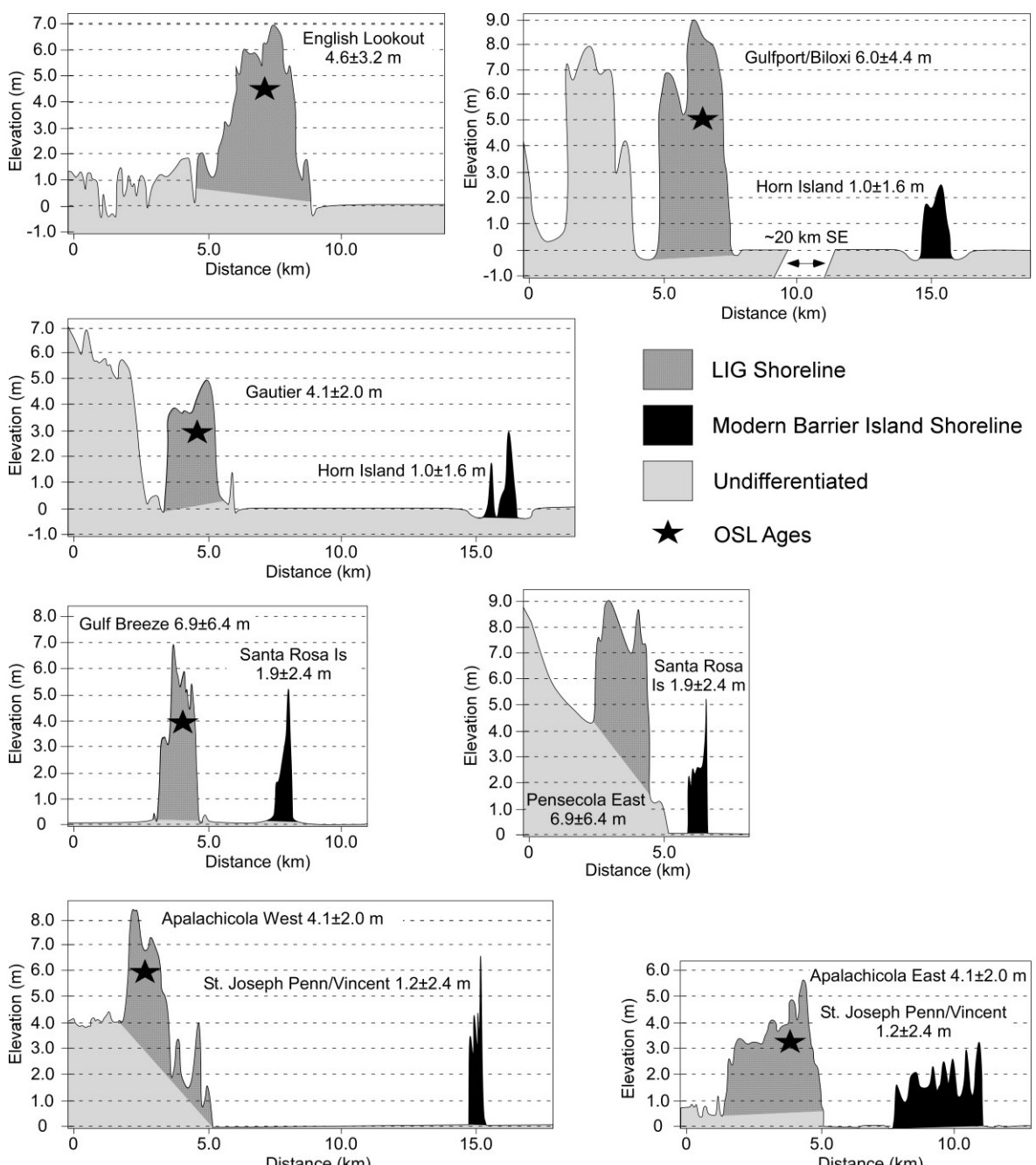

**Figure 6.** Topographic profiles through selected portions of the Mississippi-Alabama-Florida coast illustrating the elevation and width differences between the Gulfport and modern barrier islands. See Figure 5 for general locations.

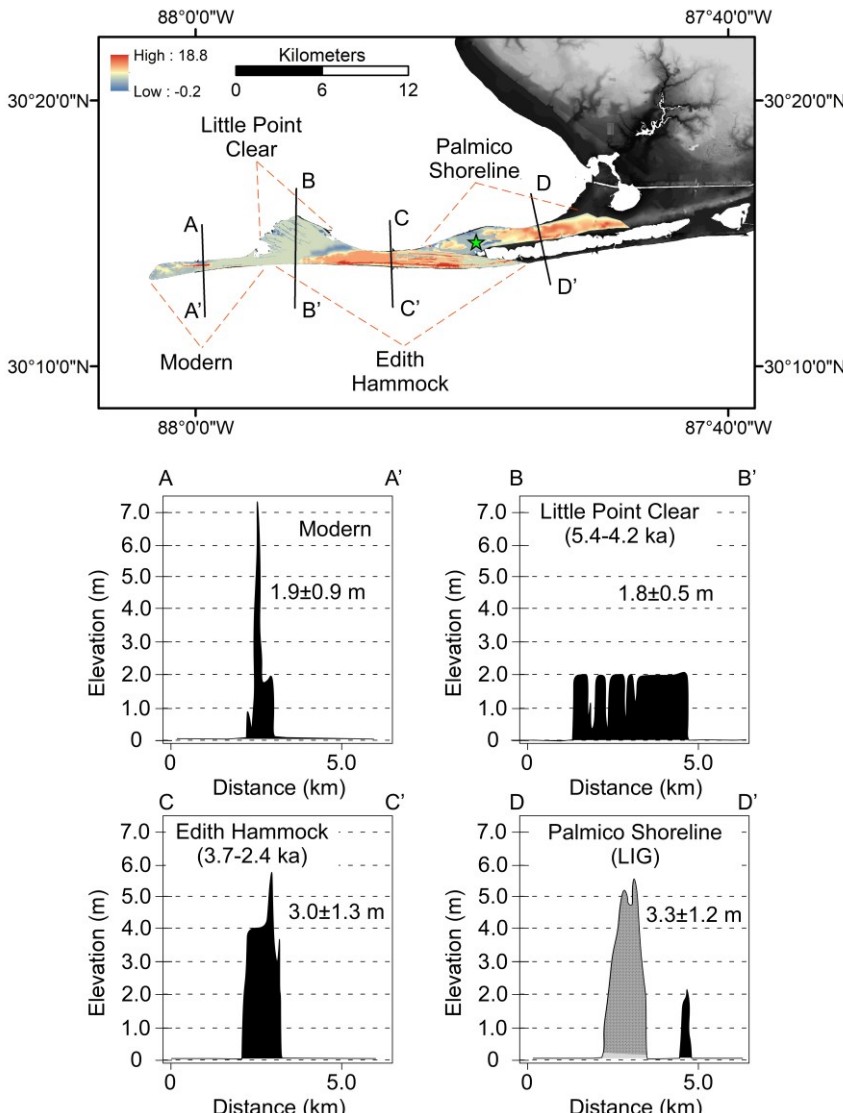

**Figure 7.** Digital elevation model and topographic profiles through the three Holocene (A-A', B-B', and C-C') and the Pamlico (LIG, D-D') shorelines. See Figure 4 for profile symbol legend.

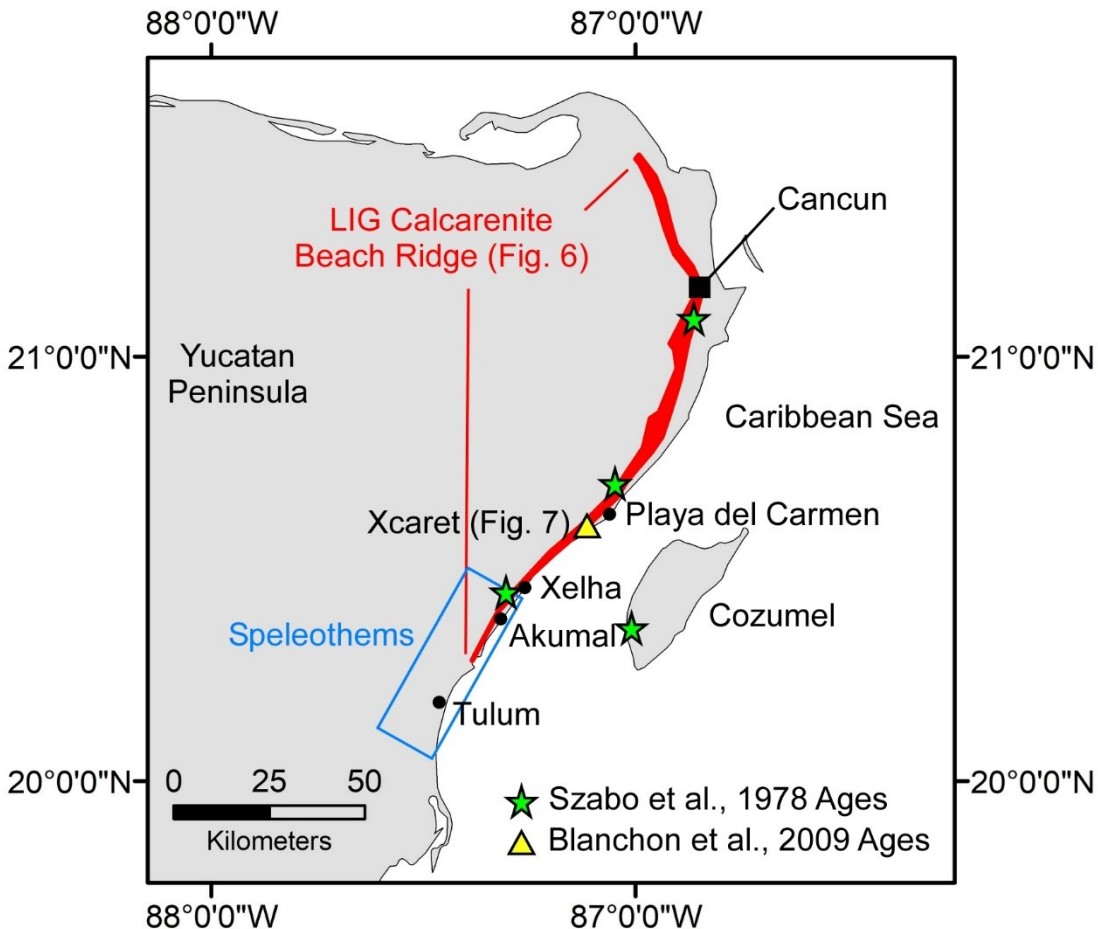

**Figure 8.** Map showing the location of LIG shoreline features of the Mexican Yucatan Peninsula.  The red strip is the location
of the LIG calcarnite beach ridge as mapped by Ward and Brady (1979), the green stars are the locations of the U-series ages
collected by Szabo et al. (1978), and the yellow triangle is the location of the Blanchon et al. (2009) study.  Also shown as a
blue box is the general location of the speleothems studied by Mosey et al. (2013).

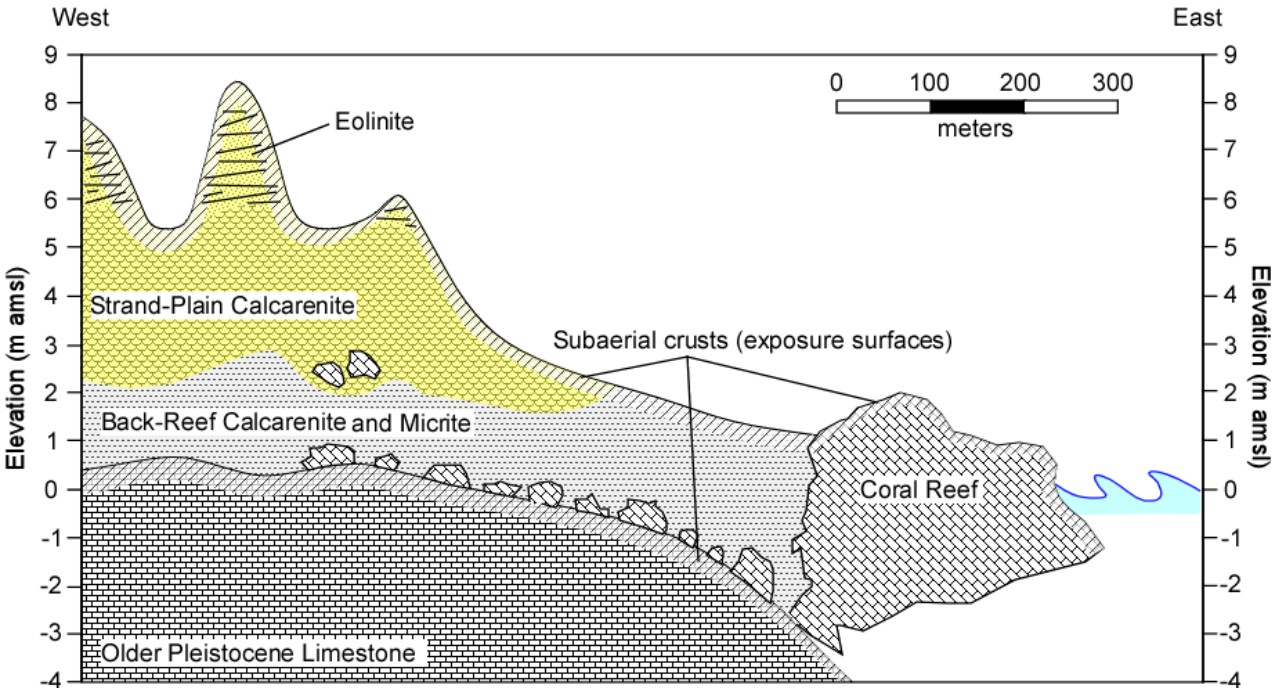

**Figure 9.** Schematic cross section through the LIG coastline of the Yucatan Peninsula of Mexico. (redrawn from Szabo et al., 1978).  See Figure 5 for a general location.

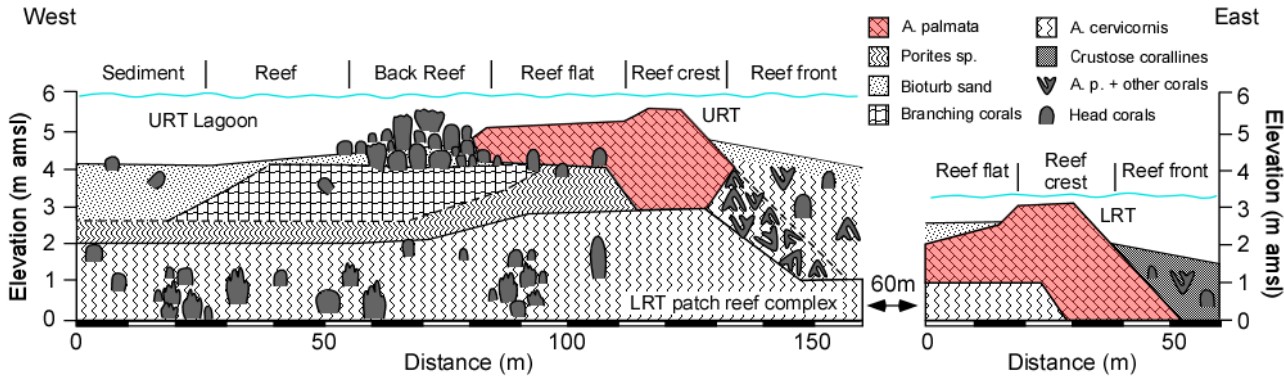

**Figure 10.** Schematic cross section through the LIG reef tracts located at Xcaret, Mexico as composed by Blanchon et al.
(2009).  LRT is lower reef tract, URT is upper reef tract, and A.p. is *Acropora palmata*.  Redrawn by permission from Blanchon et al. (2009).

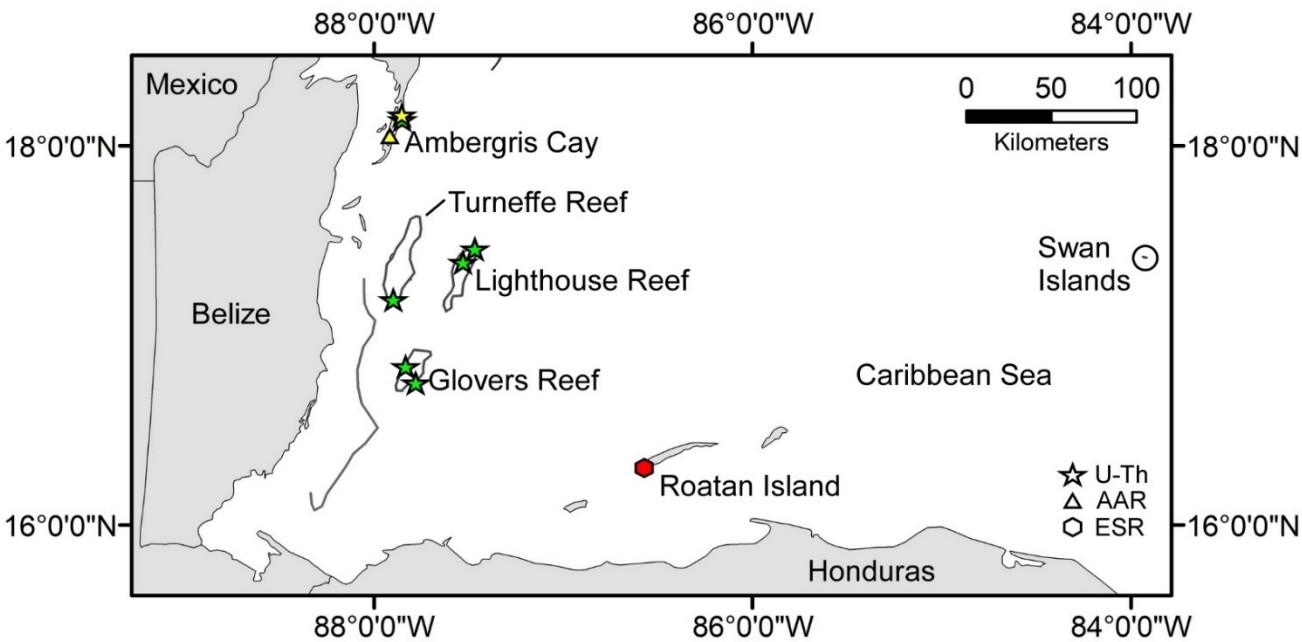

**Figure 11.** Map of the Belize and Honduras coastline showing the locations of U-Th (stars), amino-acid racemization (shown as triangles), and electron spin resonance (shown as hexagons) ages discussed in the text. Ages colored in green denote the work of Gischler et al. (2000). Ages shown in yellow denote the work of Mazzullo (2006) and ages shown in red denote the work of Cox et al. (2008)

**Table 1.** Last Interglacial Gulf of Mexico Shoreline Elevations and Relative Sea Levels

| LIG Section | Avg. Elevation (m) | 2 Stand. Dev. (m) | Modern Barrier/Shoreline | Avg. Elevation (m) | 2 Stand. Dev. (m) | Difference (LIG RSL) (m) | Error* (m) | Stratigraphic RSL (m) | Error (m) |
|---|---|---|---|---|---|---|---|---|---|
| **Northwestern Gulf of Mexico (from Simms et al., 2013)** | | | | | | | | | |
| Vidor, TX | 7.9 | 1.0 | Sabine Pass | 0.7 | 0.8 | 7.2 | 3.1 | | |
| Fannett, TX | 5.8 | 2.2 | Sabine Pass | 0.7 | 0.8 | 5.1 | 3.7 | | |
| Pine Island, TX | 2.8 | 2.2 | Bolivar Peninsula | 1.1 | 1.6 | 1.7 | 3.9 | | |
| Smith Point, TX | 2.2 | 1.8 | Bolivar Peninsula | 1.1 | 1.6 | 1.1 | 3.7 | | |
| Hoskins Island, TX | 1.0 | 0.6 | Follet's Island | 0.8 | 1.2 | 0.2 | 3.1 | | |
| Chocolate Bayou, TX | 1.2 | 0.8 | Follet's Island | 0.8 | 1.2 | 0.4 | 3.2 | | |
| Port O'Connor, TX | 2.8 | 1.8 | Matagorda Island | 1.3 | 2.0 | 1.5 | 3.9 | | |
| Aransas, TX | 3.3 | 2.4 | Saint Joseph's Island | 1.6 | 2.2 | 1.7 | 4.3 | | |
| Live Oak, TX | 4.7 | 3.6 | Mustang Island | 1.9 | 1.8 | 2.8 | 4.9 | 2.75 | 2.0 |
| Encinal, TX | 5.3 | 2.6 | North Padre Island | 1.7 | 2.0 | 3.6 | 4.3 | | |
| **Northeastern Gulf of Mexico (this study)** | | | | | | | | | |
| English Lookout (Bay Saint Louis), MS | 4.6 | 3.2 | Horn Island | 1.0 | 1.6 | 3.6 | 4.6 | | |
| Biloxi/Gulfport, MS | 6.0 | 4.4 | Horn Island | 1.0 | 1.6 | 5.0 | 5.5 | | |
| Gautier, MS | 4.1 | 2.0 | Horn Island | 1.0 | 1.6 | 3.1 | 3.8 | | |
| Pamlico (Morgan Peninsula), AL | 3.3 | 2.4 | Morgan Peninsula | 2.5 | 2.4 | 0.9 | 4.4 | 2.0 | 1.1 |
| Gulf Breeze, FL | 6.9 | 6.4 | Santa Rosa Island | 1.9 | 2.4 | 5.0 | 7.4 | | |
| Apalachicola, FL | 4.1 | 2.0 | Saint Vincent Island/Saint Joseph Peninsula | 1.2 | 2.4 | 2.9 | 4.2 | 2.75 | 1.0 |

*Also included in the root square of the sums is a +/-2.0 m error to account for errors in the DEM

**Table 1.** Last Interglacial Gulf of Mexico Shoreline Elevations and Relative Sea Levels