# Peer review of "Last interglacial sea levels within the Gulf of Mexico and northwestern Caribbean Sea"

_Earth System Science Data, 2020_

## Short Comment (SC1) · 29 Oct 2020

In an attempt to improve our scientific approach, I wish to make the following comment regarding the data reported for the northern Gulf of Mexico coast. These are listed here and published in Simms et al. (2013). Alexander Simms state that a number of sedimentary features indicate "an old barrier-beach system" that would maintain the topography of "old beach ridges". For estimating the LIG elevation of the Gulfport barrier the modern analogue approach was used: The elevation "was determined by subtracting the average elevations of the closest modern barrier islands from the average elevations of the five segments of the Gulfport Shoreline dated". "LIG beach ridges were 4-5 m in height while the modern beach ridges were 2-3 m in height". "We thus assign a modern analogue value of 2.5±1.0 m and a LIG elevation of 4.5±0.5

m for the LIG beach ridge elevation." "This suggests a LIG RSL of +2.0±1.1 m". For the northwestern coast OSL samples were collected from "the core of the barrier", i.e. from sand in core depths of 140 cm, 250 cm and 380 cm. These ages were "obtained for the Ingleside shoreline". I think this approach delivers inaccurate sea-level index points for the following reasons:

1 - The height of a barrier is controlled by local parameters such as sand supply, accommodation space, wind regime and high-energy events. Instead, the beach/upper-shoreface facies overlying the uppermost flooding surface should be the sea-level indicator. The surface marks the latest rollover or overstepping event and the overlying shoreline-related deposits provide the indicative meaning (IM). In the Holocene barrier the uppermost flooding surface is at ca 160 cm depth in the barrier where it truncates back-barrier and fluvial-deltaic deposits (Odezulu et al., 2018, in: Barrier dynamics and response to changing climate. Springer, p.147ff). 2 - The modern analogue is a great tool for first-order approximating the usefulness of a sea-level indicator, but it is mostly not sufficient for quantifying the LIG shoreline position. Blum et al. (2008; Geology) show how post-glacial sediment re-distribution impact on the lower Mississippi valley and this, in turn, should influence the sediment supply to the Ingleside shore during the Holocene. 3 - There is no evidence that the OSL ages were obtained from the sea-level indicator, i.e the beach facies. 4 – There is an unfortunate mix of terms: beach ridge seems to be used synonymously to shoreline, shoreline synonymously to barrier and barrier island synonymously to barrier. However, each of these coastal features have a different IM (e.g. Rovere et al., 2016; QSR, for beach ridge) and a shoreline is an undatable theoretical line.

The barrier complex is a high-quality indicator with a well-defined IM and IR and, because it occurs on a microtidal coast, IR/2 is small. This should be explored for Gulfport and Ingleside.

---

## Referee Comment (RC1) · Paul Blanchon (Referee) · 3 Feb 2021

The author has made a meta-analysis of 62 papers investigating RSL indicators in fossil beach-dune systems and coral reefs with LIG U-series or OSL ages around the Gulf of Mexico and the eastern Yucatan Peninsula. In general the scholarship is good and the manuscript reads well. However there are issues with consistency in determining valid estimates of RSL, methodological problems, and unstated assumptions that need addressing, as outlined below.

For the LIG beach-dune systems rimming the Gulf of Mexico, he finds that few of the studies have quantitative estimates of RSL and attempts to piece together RSL indicators based on a comparison between the average elevation of modern beach-dune

systems and the fossil systems (drawing from his own work in the area). For the NE Gulf he estimates that RSL was +1 to +5 m above present, and for the NW Gulf +2 to +2.75. The stated errors on these estimates however are large, and in some cases exceed the amplitude of the estimated highstand.

In addition to this uncertainty, there are several unstated assumptions in this analysis which I think could be addressed. First, comparing the elevation on modern beach-dune systems directly with their LIG counterparts ignores any subsequent erosion, both during the initial SL downdraw when these deposits first became inactive but were still composed of mobile sands, and later when they became part of the inshore during the 125 m lowstand. In active modern systems there is a balance between deposition and erosion, but once they became inactive, the dominant process would be erosion. Similarly the second assumption, that the elevation variability of modern beach-dune systems is representative of the LIG systems downplays the differences between transgressive vs regressive systems. For example, transgressive systems tend to be sediment starved compared to (forced) regressive systems where a slight fall in SL can expose large unlithified sediment sources and thus contribute to higher sediment flux.

For the LIG beach-dune systems running along the NE Yucatan coast, the author ignores the elevation of the beach ridge (and does not attempt to compare it with modern systems in the area) and focuses instead on the more reliable boundary between the cross-bedded beach facies and fore/back shore dunes which is reported by Ward and Brady (1979) to be +4.8 m. However the interpretation of the RSL highstand by these authors is +5 to +6 m, which is inexplicably accepted and a RSL of 5.5 ±1.5 m is assigned. There is no analysis of stratigraphic evidence to support this interpretation or it's uncertainty. Is this beach-dune system representative of the RSL highstand, or its subsequent regressive stage?

This is followed by a strange section on dated corals found in lagoonal units associated with the beach-dune systems. (Again the significance of the underlying/adjacent

lagoonal unit is unclear in terms of transgressive vs regressive stage). The author attempts to use dated, non-depth-specific coral genera (Montastrea, which is now Orbicella) to constrain the RSL highstand, following the protocol in Hibbert et al 2016. The author states that using Hibberts strict interpretation a Montastrea (species unstated) at +2 m gives a RSL interpretation of 11.7 +8.6/-7.3 m. This makes no sense. You cannot precisely constrain the RSL highstand using an unidentified coral, with a low-precision age, and a large habitat range. And you cannot say that an in-place coral found at +2 m could have grown 7.3 m deeper! The only option is to determine their consistency with respect to the most reliable RSL indicators (like an intertidal beach or reef crest). This is a problem with the Hibbert et al (2016) protocol and should be addressed in the section on 'uncertainty and data quality'.

For the LIG reef systems in the same area, the author details the elevation of reliable RSL indicators such as reef crests, before dismissing these in favor of individual corals. This time the coral species is the depth-restricted reef-crest coral A. palmata, which is only a reliable RSL indicator when found in a monospecific assemblage. Using the coral's total depth range (as suggested by Hibbert et al 2016) clearly dilutes its utility as a RSL indicator. So instead of using the most precise indicator, the reef crest itself, the elevation of in-situ corals from the 0-5 m depth zone are used to determine the RSL highstand, giving a +6.4 (+1.2/-7.9) value for an in-situ coral at 4.9 m. Again this makes no sense. You cannot claim that an in-situ coral found at +4.9 m actually grew 7.9 m below this level. The only thing you can say is it grew at a maximum of 1.2 m below SL. When evidence of a lower reef-crest unit is assessed, the level of the reef crest is used as the RSL indicator, not the elevation of its corals. This is correct, but completely the reverse of what was accepted for the highstand reef. This inconsistency is the problem.

For LIG reefs from Belize, the same problems occur with the age and elevation of individual corals being used to define RSL highstand estimates that are significantly below those in the Yucatan. These may be a result of subsidence, as the author suggests, but it could be that reef development during the LIG occurred at a lower stand of SL before

the highstand was attained, and that the other non-reefal deposits developed further inland along the unstudied coast of Belize. (i.e., that reefal deposits might equally be transgressive systems and do not represent highstand units). Clearly without a precise and reliable chronology different stages of development cannot be identified. And regardless of what geochronologists claim, the present system of correcting LIG ages for open-system behaviour has yet to provide a well-constrained SL reconstruction, or even stratigraphic consistency between and within sedimentary units.

Details: Line 383: the tidal range stated by Blanchon et al 2009 and Blanchon 2010 is 0.3 m with any data point having an uncertainty of $\pm$0.15 m.

---

## Referee Comment (RC2) · Michael O'Leary (Referee) · 8 Feb 2021

The Author presents a detailed summary of last interglacial shorelines around the Gulf of Mexico.

It is clear that many of the studies reviewed for this paper, particularly those of the US Gulf coast did not contain or reported low quality data, specifically palaeoshoreline age control and elevations, and therefore challenging to extract meaningful data to include into the WALIS database.

What was missing is a more detailed description of shoreline/coastal geomorphology, particularly along the siliciclastic dominated US Gulf coast. It is important when comparing the elevations of modern and LIG shoreline elevations, on make inferences on last interglacial sea level elevations, that the formations are comparable, i.e., comparing the elevations of a LIG barrier island and modern barrier island is reasonable, but directly comparing a LIG strand plain and modern barrier island is not. It was not made clear in the manuscript that when comparing the average elevations of modern and LIG shorelines that you are comparing like for like. It was also not made clear how the average elevations of modern and LIG coastal geomorphic features were calculated.

Also missing from the manuscript is any detailed description of the sea level indicators used in the calculations of LIG sea level's and their indicative ranges. The was no mention of the influence of GIA along the Gulf of Mexico coast and how this along with neotectonics may result in smaller or larger differences in the relative height difference between modern and and LIG shorelines.

While the figures are fine it would be better if there was high resolution DEM imagery or even a topographic profile of the modern and adjacent LIG shorelines so the reader is able to make an assessment on how similar of different they are geomorphically, and whether it is a simple as comparing the relative height difference between the two shorelines or if they are sufficiently different having formed under different metocean/sediment supply regimes that a more nuanced analysis of the indicative range of these sea level indicators is made.

I have made additional comments in the attached PDF file

Please also note the supplement to this comment:
https://essd.copernicus.org/preprints/essd-2020-253/essd-2020-253-RC2-supplement.pdf
* * *
[Figure]

**Supplement:**

[revised manuscript text omitted]

---

## Author Comment (AC1) · 23 Feb 2021

Short Comment: In an attempt to improve our scientific approach, I wish to make the following comment regarding the data reported for the northern Gulf of Mexico coast. These are listed here and published in Simms et al. (2013). Alexander Simms state that a number of sedimentary features indicate "an old barrier-beach system" that would maintain the topography of "old beach ridges".

My Response: I thank the reader for taking interest in my work and providing some insightful comments. I am sorry if I was misleading. What I meant is that some of the old "barrier islands" preserve "beach ridges" ontop of them.

Short Comment: For estimating the LIG elevation of the Gulfport barrier the modern

analogue approach was used: The elevation "was determined by subtracting the average elevations of the closest modern barrier islands from the average elevations of the five segments of the Gulfport Shoreline dated". "LIG beach ridges were 4-5 m in height while the modern beach ridges were 2-3 m in height". "We thus assign a modern analogue value of 2.5±1.0 m and a LIG elevation of 4.5±0.5 m for the LIG beach ridge elevation." "This suggests a LIG RSL of +2.0±1.1 m".

My Response: We are using two different approaches. The text above is specifically for the Morgan Peninsula case, where true "beach ridges" of both LIG and modern have been identified. Thus unlike the estimates from Simms et al. (2013), the modern analogue used is a beach ridge rather than a "Barrier Island." Antedontly, we are quoting Rodriguez and Meyer (2006) who collected GPR over both the LIG beach ridges as well as three phases of Holocene beach ridges. They were using specific stratigraphic contacts (the contact between eolian and foreshore) for their sea-level comparisons. For the rest of our Ingleside and Gulfport comparisons we compared the elevations of the entire barrier island features.

Short Comment: For the northwestern coast OSL samples were collected from "the core of the barrier", i.e. from sand in core depths of 140 cm, 250 cm and 380 cm. These ages were "obtained for the Ingleside shoreline". I think this approach delivers inaccurate sea-level index points for the following reasons:

My Response: This was used for the ages not the elevations of the LIG. The importance to the "core" of the feature in this context is that we are dating the actual Pleistocene feature and not the reworked upper sandy veneer of the feature. Unfortunately due to the coring apparatus used (a Geoprobe) very little sedimentary structures were preserved due to the percussion of the instrument. In addition, some of the recovered cores were water saturated and thus largerly homogenized for each individual core section recovered (<100 cm).

Short Comment: 1 - The height of a barrier is controlled by local parameters such as

sand supply, accommodation space, wind regime and high-energy events. Instead, the beach/uppershoreface facies overlying the uppermost flooding surface should be the sea-level indicator. The surface marks the latest rollover or overstepping event and the overlying shoreline-related deposits provide the indicative meaning (IM). In the Holocene barrier the uppermost flooding surface is at ca 160 cm depth in the barrier where it truncates back-barrier and fluvial-deltaic deposits (Odezulu et al., 2018, in: Barrier dynamics and response to changing climate. Springer, p.147ff).

My Response: I agree but the identification of specific beach facies such as upper shoreface, foreshore, or backshore are only described for a couple locations (e.g. Paine, 1993; Burdette et al., 2013). When available, we have utilized those with their indicative meanings. Unfortunately for the rest of the mapped and in many places dated Ingleside and Gulfport features, no such detailed stratigraphic information or surveyed elevations are available. Thus I was left with the average elevation of the barrier islands approach.

Short Comment: 2 - The modern analogue is a great tool for first-order approximating the usefulness of a sea-level indicator, but it is mostly not sufficient for quantifying the LIG shoreline position. Blum et al. (2008; Geology) show how post-glacial sediment re-distribution impact on the lower Mississippi valley and this, in turn, should influence the sediment supply to the Ingleside shore during the Holocene.

My response: You are correct, but that is all that is available. I have highlighted these assumptions and applied a liberal error along with noting that within the "Quality of the data" portion of the WALIS database. The poorly constrained data are better than no data as long as its limitations are acknowledged.

Short Comment: 3 - There is no evidence that the OSL ages were obtained from the sea-level indicator, i.e the beach facies. 4 – There is an unfortunate mix of terms: beach ridge seems to be used synonymously to shoreline, shoreline synonymously to barrier and barrier island synonymously to barrier. However, each of these coastal
features have a different IM (e.g. Rovere et al., 2016; QSR, for beach ridge) and a shoreline is an undatable theoretical line.

My response: The ages were obtained from the sand deposits of the Ingleside feature but yes due to liquefaction of the cores we were not able to deduce any detailed sedimentology and thus facies. As for the mix of terms, thank you for the comment. I have tried to be clear what I am referring to. I use "beach ridge" to describe the specific geomorphic ridge features (sensu Otvos, 2020) developed ontop of the modern barrier islands or LIG features (See Figure 2). We use "shoreline" as a general term for the Ingleside and Gulfport features as mapped with no concept of what type of specific landform (e.g. barrier island, mainland-attached beach, strandplain, etc.) they represent other than old sandy, presumably shallow marine features. We use "barrier island" when interpreting the landforms as official geomorphic features of a sandy narrow and long depositional features separating the open ocean from some sort of backbarrier environment such as a lagoon or coastal marsh. We have done away with the term "barrier", it was originally used to suggest that the LIG features may not have been true "barrier island" but potentially shore attached beaches or the "mainland barriers" of Otvos (2020).

Short Comment: The barrier complex is a high-quality indicator with a well-defined IM and IR and, because it occurs on a microtidal coast, IR/2 is small. This should be explored for Gulfport and Ingleside.

My response: That is what I did – the average elevation of the LIG barrier island (or complex) minus the elevation of the modern barrier island (or complex). We have tried to be more clear about that ("Assuming the Ingleside was a LIG barrier island (Price, 1933; Paine, 1993) similar to the modern barrier islands of the Texas Gulf Coast, which is still a matter of discussion (Otvos, 2018, 2020), Simms et al. (2013) subtracted the average elevation of the closest modern equivalent barrier island from the elevation of each of the Ingleside shoreline segments of the Texas Coast.")

---

## Author Comment (AC2) · 23 Feb 2021

*Reviewer Comment: The author has made a meta-analysis of 62 papers investigating RSL indicators in fossil beach-dune systems and coral reefs with LIG U-series or OSL ages around the Gulf of Mexico and the eastern Yucatan Peninsula. In general the scholarship is good and the manuscript reads well. However there are issues with consistency in determining valid estimates of RSL, methodological problems, and unstated assumptions that need addressing, as outlined below. For the LIG beach-dune systems rimming the Gulf of Mexico, he finds that few of the studies have quantitative estimates of RSL and attempts to piece together RSL indicators based on a comparison between the average elevation of modern beach-dune systems and the fossil systems (drawing from his own work in the area). For the NE Gulf he estimates that

RSL was +1 to +5 m above present, and for the NW Gulf +2 to +2.75. The stated errors on these estimates however are large, and in some cases exceed the amplitude of the estimated highstand. In addition to this uncertainty, there are several unstated assumptions in this analysis which I think could be addressed. First, comparing the elevation on modern beachdune systems directly with their LIG counterparts ignores any subsequent erosion, both during the initial SL downdraw when these deposits first became inactive but were still composed of mobile sands, and later when they became part of the inshore during the 125 m lowstand. In active modern systems there is a balance between deposition and erosion, but once they became inactive, the dominant process would be erosion. Similarly the second assumption, that the elevation variability of modern beach-dune systems is representative of the LIG systems downplays the differences between transgressive vs regressive systems. For example, transgressive systems tend to be sediment starved compared to (forced) regressive systems where a slight fall in SL can expose large unlithified sediment sources and thus contribute to higher sediment flux.

**My response: I thank the reviewer for their careful reading of the manuscript and many insightful and helpful comments. Yes, I have made many assumptions in comparing the average elevation of the modern and LIG paleo barrier islands. I have added a paragraph acknowledging more explicitly these assumptions including those brought up by the reviewers. Concerning the transgressive versus regressive nature of the barrier islands, I tried to compare similar features to similar features. In general the modern aggradational and progradational barrier islands are wider than their transgressive counterparts (Anderson et al., 2014; Otvos and Carter, 2013). For example compare the width of the transgressive Follet's Island with the aggradational Mustang Island (Fig. 4). The Ingleside segments of the South Texas coast are wider with preserved beach ridges and thus likely represent regressive barrier islands. But so are their modern equivalents (see new Figure 4). Conversely, the lower elevation and thinner segments of the Ingleside near the Brazos-Colorado River Delta (e.g. Hoskins and Chocolate Bayou) are similar in width to their analogue east-Texas barrier of Folletts

Island (see new Figure 4). With the exception of the Vidor segment this holds true for most of our analogue pairs (See new Figure 4) for the Texas LIG shoreline segments. This does not however, hold true for the eastern Gulf of Mexico LIG features. We have acknowledged that (lines 500-505) and hope the larger error bars applied to the estimates derived this way account for those differences. I have doubled the error estimates from the Simms et al. (2013) study to help account for the larger uncertainty given our assumptions.

*Reviewer Comment: For the LIG beach-dune systems running along the NE Yucatan coast, the author ignores the elevation of the beach ridge (and does not attempt to compare it with modern systems in the area) and focuses instead on the more reliable boundary between the cross-bedded beach facies and fore/back shore dunes which is reported by Ward and Brady (1979) to be +4.8 m. However the interpretation of the RSL highstand by these authors is +5 to +6 m, which is inexplicably accepted and a RSL of 5.5 ±1.5 m is assigned. There is no analysis of stratigraphic evidence to support this interpretation or it's uncertainty. Is this beach-dune system representative of the RSL highstand, or its subsequent regressive stage?

**My response: Thank you for pointing out where my analysis could be better. I have revisited the estimates based on the beach ridges and used the contact between the upper shoreface/foreshore instead. Using the IMCalc program (Lorscheid and Rovere, 2019), I approximate the "breaking depth' of the waves and apply that value when assigning an indicative meaning to the stratigraphic contact. We do assume that the beach ridges and their related underlying shoreface/foreshore deposits represent the highstand and I added a statement about that assumption.

*Reviewer Comment: This is followed by a strange section on dated corals found in lagoonal units associated with the beach-dune systems. (Again the significance of the underlying/adjacent lagoonal unit is unclear in terms of transgressive vs regressive stage). The author attempts to use dated, non-depth-specific coral genera (Montastrea, which is now Orbicella) to constrain the RSL highstand, following the protocol in

Hibbert et al 2016. The author states that using Hibberts strict interpretation a Montastrea (species unstated) at +2 m gives a RSL interpretation of 11.7 +8.6/-7.3 m. This makes no sense. You cannot precisely constrain the RSL highstand using an unidentified coral, with a low-precision age, and a large habitat range. And you cannot say that an in-place coral found at +2 m could have grown 7.3 m deeper! The only option is to determine their consistency with respect to the most reliable RSL indicators (like an intertidal beach or reef crest). This is a problem with the Hibbert et al (2016) protocol and should be addressed in the section on 'uncertainty and data quality'.

**My Reponse: I removed the discussion related to the Hibberts et al. (2016) applications for where other data is available (everyone but Belize) and simply use the ages as limiting data, which agrees with the revised overlying beach ridge-based estimates. We included a statement about the uncertainty introduced by using the Hibbert et al. (2016) protocol in the section on "uncertainty and data quality."

*Reviewer Comment: For the LIG reef systems in the same area, the author details the elevation of reliable RSL indicators such as reef crests, before dismissing these in favor of individual corals. This time the coral species is the depth-restricted reef-crest coral A. palmata, which is only a reliable RSL indicator when found in a monospecific assemblage. Using the coral's total depth range (as suggested by Hibbert et al 2016) clearly dilutes its utility as a RSL indicator. So instead of using the most precise indicator, the reef crest itself, the elevation of in-situ corals from the 0-5 m depth zone are used to determine the RSL highstand, giving a +6.4 (+1.2/-7.9) value for an in-situ coral at 4.9 m. Again this makes no sense. You cannot claim that an in-situ coral found at +4.9 m actually grew 7.9 m below this level. The only thing you can say is it grew at a maximum of 1.2 m below SL. When evidence of a lower reef-crest unit is assessed, the level of the reef crest is used as the RSL indicator, not the elevation of its corals. This is correct, but completely the reverse of what was accepted for the highstand reef. This inconsistency is the problem.

**My response: I thank the reviewer for pointing out the shortcomings of simply applying the results of the Hibbert et al. (2018) zonation of individual coral species to the well-documented reef facies at Xcarat. Thus I have changed the LIG RSL based on the Xcaret data to fully utilize the stratigraphic information at the site. I struggled to find a reference citing specifically the reef crest elevation for Carribean corals but did find one for the reef flat. A study by Cubit et al. (1986) found that the reef flat forms 6 cm below MLLW. I added to that $\frac{1}{2}$ the tidal range (15 cm) to arrive at the reef flats indicative meaning of 20 cm below mean sea level. Thus RSL at the LIG based on the higher reef tract at Xcaret is +5.7+/-0.2 m, the error being the root sum of the squares of the indicative meaning (1/2 the tidal range) and the measurement error. This indicative meaning assignment for the reef flat places the related reef crest at about mean sea level, which is what I think the reviewer was alluding to for its indicative meaning.

*Reviewer Comment: For LIG reefs from Belize, the same problems occur with the age and elevation of individual corals being used to define RSL highstand estimates that are significantly below those in the Yucatan. These may be a result of subsidence, as the author suggests, but it could be that reef development during the LIG occurred at a lower stand of SL before the highstand was attained, and that the other non-reefal deposits developed further inland along the unstudied coast of Belize. (i.e., that reefal deposits might equally be transgressive systems and do not represent highstand units). Clearly without a precise and reliable chronology different stages of development cannot be identified. And regardless of what geochronologists claim, the present system of correcting LIG ages for open-system behaviour has yet to provide a well-constrained SL reconstruction, or even stratigraphic consistency between and within sedimentary units.

**My response: Yes, I agree that the analysis on the Belize reefs is also fraught with the limitations of the Hibberts et al. (2016) approach (which I have now acknowledged in the section as already suggested). Therefore for the Ambergis Cay site, which Mazzullo (2006) describes as a reef flat facies, I used the same approach as with the Xcaret data. However, as the elevation datums and facies architecture are not as well characterized

as they are at Xcaret, I provide a larger error of $\pm 1.0$ m. However, the other Belize reefs are not as well described in terms of their sedimentary characteristics and indicative meaning (we added a statement to that fact on lines 387-388). Thus I was only left with two options for my interpretations of those deposits: a generic, shallow marine reef with an arbitrary error assignment or use the approach of Hibberts et al. (2016). I favor the latter. Nevertheless, I point out these uncertainties including the reiteration of the point that the Belize reefs might not necessarily represent the highest MIS5e deposits but could represent earlier (transgression leading up to MIS5e) or later reefs (e.g. post MIS5e) in line 418.

*Reviewer Comment: Details: Line 383: the tidal range stated by Blanchon et al 2009 and Blanchon 2010 is 0.3 m with any data point having an uncertainty of $\pm 0.15$ m.

**My response: I changed this, thank you.

---

## Author Comment (AC3) · 23 Feb 2021

*Reviewer Comment: The Author presents a detailed summary of last interglacial shorelines around the Gulf of Mexico. It is clear that many of the studies reviewed for this paper, particularly those of the US Gulf coast did not contain or reported low quality data, specifically palaeoshoreline age control and elevations, and therefore challenging to extract meaningful data to include into the WALIS database. What was missing is a more detailed description of shoreline/coastal geomorphology, particularly along the siliciclastic dominated US Gulf coast. It is important when comparing the elevations of modern and LIG shoreline elevations, on make inferences on last interglacial sea level elevations, that the formations are comparable, i.e., comparing the elevations of a LIG barrier island and modern barrier island is reasonable, but directly comparing a LIG

strand plain and modern barrier island is not. It was not made clear in the manuscript that when comparing the average elevations of modern and LIG shorelines that you are comparing like for like. It was also not made clear how the average elevations of modern and LIG coastal geomorphic features were calculated.

**My response: Thank you for the comment. I have tried to be more explicit that we are comparing the average elevation of a LIG barrier island to a modern barrier island. In order to allow the readers to compare the similarity between the modern and LIG barrier islands we have included elevation profiles through the two features.

*Reviewer comment: Also missing from the manuscript is any detailed description of the sea level indicators used in the calculations of LIG sea level's and their indicative ranges.

**My response: I have tried to be more explicit in our indicative ranges, having added that language on lines 163, 169, 176, 184, 249, 378, 388, and 516.

*Reviewer Comment: The was no mention of the influence of GIA along the Gulf of Mexico coast and how this along with neotectonics may result in smaller or larger differences in the relative height difference between modern and and LIG shorelines.

**My response: The Simms et al. (2013) study did include GIA, I added a statement to make that clear. As for neotectonics, growth faults are present but the actively mapped growth faults are seaward of the LIG shoreline – I have added a sentence (line 74) to address that while providing more background for the general area as requested by a reviewer.

*Reviewer Comment: While the figures are fine it would be better if there was high resolution DEM imagery or even a topographic profile of the modern and adjacent LIG shorelines so the reader is able to make an assessment on how similar of different they are geomorphically, and whether it is a simple as comparing the relative height difference between the two shorelines or if they are sufficiently different having formed

under different metocean/sediment supply regimes that a more nuanced analysis of the indicative range of these sea level indicators is made.

**My response: Thank you for the comment, we have added figures (4, 6, and 7) showing topographic profiles through the features to allow for a comparison between the LIG "barrier islands" and their analogous modern barrier islands.

*Reviewer Comment: I have made additional comments in the attached PDF file.

**My response: Most of the comments on the annotated PDF were minor editorial suggestions (wording change or please include a definition for that). The following is reprinted list of the comments made on the annotated PDF that were more substantial than a simply expansion or grammar correction:

*Reviewer Comment: Line 126: "Not sure what you mean by turn-round? can this explain the volume of sediment that form this shoreline, up to 16 km wide and 30 m think, I would suspect you would need an extended interval or stable sea level to build such a massive shoreline feature"

**My response: - Rewrote as "maximum shoreline transgression" - Modern (more appropriately, the Holocene) Mustang Island is up to 30 m thick and the strain plain at Sabine Pass extends over a width of 15 km – both formed within the last 7,000 years (near the extent of the LIG highstand).

*Reviewer comment: Line 128: Have there been more recent studies:

**My response: To my knowledge there have been no new studies focused on these eastern portions of the Ingleside since the work of Otvos (1997)

*Reviewer Comment: Line 141: "How can dunes "cover" a shoreline at the time of formation, a dune and shoreline form contemporaneously?"

**My response: Today along progradational barriers dunes of less than a couple hundred years overlie beach deposits of less than a couple hundred years – e.g. Bolivar

[Figure]

Peninsula (Rodriguez et al., 2004 – JSR).

*Reviewer Comment: Line 151: "It is really not clear to me how you came to these values, you have not discussed the tidal range, how you came to conclude the indicative range of your shell horizons is limiting value of -2.5 m? If the shell layer represents a beach/shoreface then given the upper and lower tidal range will be you the upper and lower limiting elevations of the shell deposit."

**My response: - As explained in lines 144-149, we arrived at this value because equivalent facies are not found above +1 m – setting the upper limit for the deposits. They could not have been deposited in water depths more than 2.5 m as that would submerge equivalent eolian deposits (which should have been deposited above contemporaneous sea levels). This gives an indicative meaning of -0.75+/-1.75. Add that to the 2 m elevation of the highest LIG shell deposits gives a value of +2.75+/-2.0.

*Reviewer comment: Line 166: What about GIA?

**My response: Yes, Simms et al. (2013) did account for GIA differences across the Ingleside locations of Texas.

*Reviewer comment: Line 191: "I think it is entirely reasonable to use the modern beach ridge elevations as a modern analogue for the LIG elevations, but I think you need to discuss potential uncertainties in the LIG elevations, such as dissolution of carbonates grains which could lower the elevations of LIG, general deflation, you have already mentioned that they may be aeolian reworking of the tops of dunes. even different wind and have climate during the LIG can result in higher or lower dune heights. It would have been useful to measure not just the dune height but the relative height difference between the dune crest and swale for both the modern and LIG beach-barriers as this would then at least let you know if there has been a post depositional deflation."

**My response: - I think we should have been more explicit about what elevation difference Simms et al. (2013) used. It wasn't the difference in elevation between the

modern and LIG beach ridge elevations but the entire paleo barrier island and modern barrier island. That is also the approach favored by the "Short Comment" of Barbara Mauz.